# Prompt Optimization with Human Feedback

## Abstract

Large language models (LLMs) have demonstrated remarkable performances in various tasks. However, the performance of LLMs heavily depends on the input prompt, which has given rise to a number of recent works on *prompt optimization*. However, previous works often require the availability of a numeric score to assess the quality of every prompt. Unfortunately, when a human user interacts with a black-box LLM, attaining such a score is often infeasible and unreliable. Instead, it is usually significantly easier and more reliable to obtain *preference feedback* from a human user, i.e., showing the user the responses generated from a pair of prompts and asking the user which one is preferred. Therefore, in this paper, we study the problem of *prompt optimization with human feedback* (POHF), in which we aim to optimize the prompt for a black-box LLM using only human preference feedback. Drawing inspiration from dueling bandits, we design a theoretically principled strategy to select a pair of prompts to query for preference feedback in every iteration, and hence introduce our algorithm named *automated POHF* (APOHF). We apply our APOHF algorithm to various tasks, including optimizing user instructions, prompt optimization for text-to-image generative models, and response optimization with human feedback (i.e., further refining the response using a variant of our APOHF). The results demonstrate that our APOHF can efficiently find a good prompt using a small number of preference feedback instances.

## 1 Introduction

Large language models (LLMs) have shown impressive performances in a variety of tasks (Google, 2023; OpenAI, 2023). However, the performances of LLMs are significantly dependent on the *prompt* given to them (Zhou et al., 2023). Unfortunately, finding the best prompt for an LLM to perform a task is often challenging, especially considering that the most powerful LLMs nowadays are often *black-box* models to which only API access is available (OpenAI, 2023). This challenge has given rise to a number of recent works on *prompt optimization* for black-box LLMs, which aim to efficiently find the best prompt for a black-box LLM (Chen et al., 2023; Lin et al., 2024; Zhou et al., 2023). These works have shown that prompt optimization can dramatically improve the performances of black-box LLMs in various tasks. However, these works often impose a potentially unrealistic requirement on the tasks: *They usually require access to a numeric score to evaluate the performance of every prompt*. This significantly limits their practicality in real-world use cases.

Specifically, some works on prompt optimization have assumed the availability of a validation set, which can be used to evaluate (the response generated from) a candidate prompt (Chen et al., 2023; Hu et al., 2024; Lin et al., 2024). Meanwhile, other works have used a separate LLM (often referred to as the scorer LLM) to provide a score indicating the efficacy of (the response produced by) a prompt (Yang et al., 2024; Zhou et al., 2023). However, *when a human user directly interacts with a black-box LLM to perform a task* (i.e., the most common use cases of LLMs nowadays), these methods to obtain a score are often unrealistic. This is because in such use cases, a validation set is usually unavailable and the scorer LLM is unlikely to provide an accurate assessment of a prompt for the task the user has in mind. Therefore, these previous prompt optimization methods are inapplicable for such use cases. In addition, directly asking a user for a numeric score to assess (the response generated by) a candidate prompt is usually infeasible and unreliable (Yue et al., 2012). Instead, a human user is often significantly more willing to and reliable at providing *preference feedback*, i.e., examining the responses generated by a pair of prompts and indicating which one is preferred (Yue et al., 2012). This naturally begs the question: **Can we achieve prompt optimization using only**

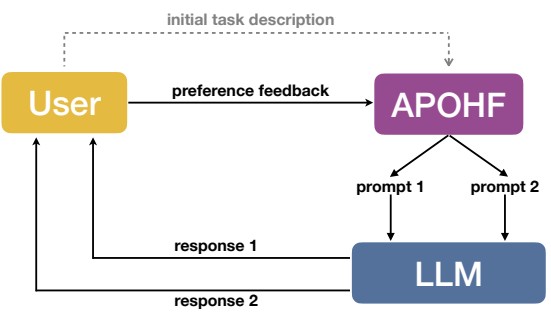

Figure 1: Illustration of our automated prompt optimization with human feedback (`APOHF`).

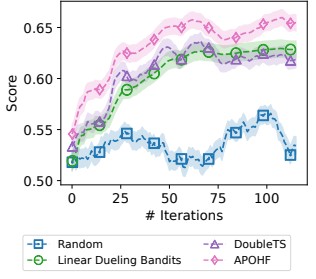

Figure 2: Latent scores of different methods in user instruction optimization, averaged over 30 tasks (Sec. 4.1).

**human preference feedback?** In this work, we tackle this important problem, which we refer to as *prompt optimization with human feedback* (POHF).

The significance of POHF can also be highlighted by drawing an analogy to *reinforcement learning with human feedback* (RLHF) (Ziegler et al., 2019). RLHF, as well as its variants such as direct preference optimization (Rafailov et al., 2024), uses a dataset of human preference feedback to fine-tune the parameters of an LLM in order to align the LLM with human values (Rafailov et al., 2024). The tremendous success of RLHF is evidence of the advantage of using human preference feedback to adapt LLMs. While RLHF has relied on fine-tuning the model parameters to adapt the response of an LLM (to align with human values), our POHF aims to *use prompt optimization to adapt the response of an LLM to perform a task for a human*. Interestingly, our algorithm for POHF can be extended to further refine the response of an LLM through *response optimization with human feedback* (Sec. 4.3). Specifically, for every received prompt, we can use the LLM to generate a large pool of responses and then strategically select a pair of responses from the pool to query for user preference feedback (Dwaracherla et al., 2024). Our goal here is to find the best response for every given prompt while using only human preference feedback. This can be useful in applications where we do not have the flexibility to choose the prompt, but can sample a large number of responses from the LLM. For example, it may be adopted by an LLM provider to further refine its response to user prompts while only collecting user preference feedback.

Similar to RLHF, in our POHF, it is of paramount importance to find a good prompt *using a small number of human feedback instances*. This is because collecting human feedback can usually be expensive and time-consuming. To achieve this, inspired by Lin et al. (2024), we adopt the embedding from a pre-trained language model (Reimers, 2019) as the continuous representation of the prompts, and train a neural network (NN), which takes the embedding as input, to predict the performance (i.e., the latent score, see Sec. 3.1) of different prompts. Based on the trained NN, we draw inspiration from *dueling bandits* (Bengs et al., 2022; Saha, 2021) and design a theoretically principled strategy to select the pair of prompts (to query for human feedback) in every iteration. Specifically, we choose the first prompt following a greedy strategy, i.e., by selecting the prompt that is predicted to have the best performance by the trained NN. Next, we select the second prompt based on the principle of upper confidence bound, which allows us to simultaneously *exploit* the performance prediction from the NN and *explore* those prompts whose performance prediction has large uncertainty. As a result of the accurate performance prediction of the NN (thanks to the expressive power of the pre-trained embedding and the NN) and our principled prompt selection strategy, our algorithm, named *Automated POHF* (`APOHF`), is able to find a good prompt using only a small number of human preference feedback instances.

Within our problem setting (illustrated in Fig. 1), our `APOHF` algorithm acts as an interface between the user and the LLM. To adopt our `APOHF` in practice, the user only needs to provide (1) an initial task description (e.g., a few input-output exemplars or an initial prompt) and subsequently (2) a series of preference feedback between pairs of responses (more details in Sec. 3.3). We adopt a number of tasks to validate the performance of our `APOHF`, including optimizating user instructions (Sec. 4.1), prompt optimization for text-to-image generative models (Sec. 4.2), and response optimization with human feedback (Sec. 4.3). In these tasks, our `APOHF` consistently achieves better performances than baseline methods, demonstrating its immense potential in real-world applications.

## 2 PROBLEM SETTING

In POHF, we aim to find a prompt $x \in \mathcal{X}$ that maximizes an unknown function $u$, which we refer to as the latent score/utility function. That is, we aim to solve the following optimization problem: $x^\star = \arg\max_{x \in \mathcal{X}} u(x)$ while only observing human *preference feedback*. In every iteration $t$, we select a pair of prompts $x_{t,1}$ and $x_{t,2}$ to obtain their corresponding LLM-generated responses and show them to the user. Then, we collect a binary observation $y_t = \mathbb{1}(x_{t,1} \succ x_{t,2})$, which is equal to 1 if the human user prefers the response from $x_{t,1}$ over that from $x_{t,2}$ and 0 otherwise. To model the preference feedback, we adopt the commonly used Bradley-Terry-Luce (BTL) model (Hunter, 2004). That is, for any pair of prompts $x_1$ and $x_2$, the probability that $x_1$ is preferred over $x_2$ is given by $\mathbb{P}(x_1 \succ x_2) = \sigma(u(x_1) - u(x_2))$, in which $\sigma(\cdot)$ denotes the logistic function: $\sigma(x) = 1/(1 + \mathrm{e}^{-x})$. The binary observation $y_t$ is then sampled from a Bernoulli distribution with probability $\mathbb{P}(x_1 \succ x_2)$. The stochastic nature of $y_t$ allows us to naturally account for the noise in human preferences between a pair of prompts. The noise may arise due to different sources of randomness, such as the randomness in the LLM-generated response for a given prompt, the variability in human decisions, among others.

Following recent works on query-efficient prompt optimization (Chen et al., 2023; Lin et al., 2024), we convert POHF into a continuous optimization problem. Specifically, for every prompt $x \in \mathcal{X}$ in the domain, we extract the embedding from a pre-trained language model as its continuous representation. In our experiments, we use Sentence-BERT (Reimers, 2019) as the embedding model. Of note, the previous works of Chen et al. (2023) and Lin et al. (2024) adopted a separate white-box LLM so that the soft prompt (i.e., a part of the input to the white-box LLM to generate the prompt) can be used as the continuous representation of the prompt. Therefore, compared to Chen et al. (2023) and Lin et al. (2024), our method of adopting the embedding from a pre-trained model removes the need for the white-box LLM, and hence significantly reduces the complexity and computational cost. To simplify notations, hereafter, we use $x$ to denote the continuous embedding of a prompt in the domain. Before the beginning of our algorithm, we use the initial task description from the user (Fig. 1) to generate the discrete domain of prompts $\mathcal{X}$, which we discuss in more detail in Sec. 3.3.

## 3 AUTOMATED PROMPT OPTIMIZATION WITH HUMAN FEEDBACK (APOHF)

**Overview of APOHF (illustrated in Fig. 1).** In every iteration $t$ of our APOHF algorithm (Algo. 1), we firstly use the current history of preference observations $\mathcal{D}_{t-1} = \{(x_{s,1}, x_{s,2}, y_s)\}_{s=1,\ldots,t-1}$ to train a neural network (NN) for score prediction (Sec. 3.1). Next, we leverage the trained NN to select the next pair of prompts $x_{t,1}$ and $x_{t,2}$ to query (Sec. 3.2). Then, the pair of prompts $x_{t,1}$ and $x_{t,2}$ are used to generate their respective responses, which are shown to the human user who gives preference feedback $y_t = \mathbb{1}(x_{t,1} \succ x_{t,2})$ (Sec. 3.3). The newly collected observation $(x_{t,1}, x_{t,2}, y_t)$ is then added to the history, which is subsequently used to train the NN for the next iteration $t + 1$.

---

**Algorithm 1** Automated Prompt Optimization with Human Feedback (APOHF)

---

1: **for** $t = 1, \ldots, T$ **do**
2:   Train NN using history $\mathcal{D}_{t-1} = \{(x_{s,1}, x_{s,2}, y_s)\}_{s=1,\ldots,t-1}$ by minimizing loss function (1)
3:   Choose the first prompt $x_{t,1}$ by maximizing the NN prediction
4:   Choose the second prompt $x_{t,2}$ by maximizing the upper confidence bound in Eq. (2)
5:   Obtain the responses from $x_{t,1}$ and $x_{t,2}$, and observe user preference: $y_t = \mathbb{1}(x_{t,1} \succ x_{t,2})$
6: Train NN using entire history, report $x_T^* = \arg\max_{x \in \{x_{s,1}, x_{s,2}\}_{s=1,\ldots,T}} h(x; \theta_T)$ as best prompt

---

### 3.1 TRAINING THE NEURAL NETWORK FOR LATENT SCORE PREDICTION

In our APOHF, we adopt an NN (more specifically, a multi-layer perceptron, or MLP) with parameters $\theta$, denoted as $h(x; \theta)$. The NN takes as input the pre-trained embedding $x$ of a prompt and predicts its latent score $u(x)$. Therefore, for a pair of prompts $x_1$ and $x_2$, we use $\sigma(h(x_1; \theta) - h(x_2; \theta))$ to model the probability that $x_1$ is preferred over $x_2$: $\mathbb{P}(x_1 \succ x_2) = \sigma(u(x_1) - u(x_2))$.

In iteration $t$, given the current history of preference observations $\mathcal{D}_{t-1} = \{(x_{s,1}, x_{s,2}, y_s)\}_{s=1,\ldots,t-1}$, we train the NN using gradient descent to minimize the following loss function:

$$\mathbf{l}_t(\theta) = -\left( \sum_{s=1}^{t-1} \left[ y_s \log \sigma\big(h(x_{s,1}; \theta) - h(x_{s,2}; \theta)\big) + (1 - y_s) \log \sigma\big(h(x_{s,2}; \theta) - h(x_{s,1}; \theta)\big) \right] \right) + \lambda \|\theta\|_2^2.$$

$$(1)$$

Recall that $y_s = \mathbb{1}(x_{s,1} \succ x_{s,2})$. Intuitively, minimizing this loss function (1) corresponds to obtaining the maximum log-likelihood estimate of the MLP parameters $\theta$ (with L2 regularization) using the preference dataset $\mathcal{D}_{t-1}$. The strong expressive power of the pre-trained embedding and the NN helps us accurately estimate the latent score function $u$, which is crucial for the strong performance of our APOHF algorithm. After the NN is trained, the resulting NN with parameters $\theta_t = \arg\min_\theta l_t(\theta)$ is used to select the pair of prompts to query in iteration $t$ (Sec. 3.2).

## 3.2 Selecting the Next Pair of Prompts

The prompt selection strategy of our APOHF is designed by drawing inspirations from the theoretically principled *linear dueling bandits* (Bengs et al., 2022; Saha, 2021). However, note that instead of using a linear model to learn the score function (Bengs et al., 2022; Saha, 2021), we adopt an NN (Sec. 3.1) to make our APOHF not only theoretically grounded but also practically effective. As we verify in Sec. 4, our APOHF substantially outperforms linear dueling bandits in all our experiments. We also provide some high-level theoretical justifications for our prompt selection strategy in App. C.

We choose **the first prompt** greedily, i.e., by selecting the one predicted to have the largest latent score using the trained NN (Sec. 3.1): $x_{t,1} = \arg\max_{x \in \mathcal{X}} h(x; \theta_t)$. Next, after the first prompt $x_{t,1}$ is selected, we choose **the second prompt** $x_{t,2}$ by maximizing an upper confidence bound:

$$x_{t,2} = \arg\max_{x \in \mathcal{X}} h(x; \theta_t) + \nu \left\| \nabla_{\theta_t} h(x; \theta_t) - \nabla_{\theta_t} h(x_{t,1}; \theta_t) \right\|_{V_{t-1}^{-1}}, \tag{2}$$

in which $V_t = \sum_{s=1}^t \varphi'_{t,s} \varphi'^\top_{t,s} + \lambda \mathbf{I}$, and $\varphi'_{t,s} = \nabla_{\theta_t} h(x_{s,1}; \theta_t) - \nabla_{\theta_t} h(x_{s,2}; \theta_t)$. Our strategy to select the second prompt (2) is able to balance the exploration-exploitation trade-off. Specifically, the first term $h(x; \theta_t)$ allows us to **exploit** the predicted score of the trained NN. Meanwhile, the second term in (2) characterizes our *uncertainty* about the score of $x$ given (a) the prompts selected in the previous iterations $\mathbf{X}_{t-1} = \{(x_{s,1}, x_{s,2})\}_{s=1,...,t-1}$ and (b) the first selected prompt $x_{t,1}$. Intuitively, a larger value of the second term (i.e., a larger uncertainty) suggests that $x$ is more different from the previously queried prompts $\mathbf{X}_{t-1}$ and the first selected prompt $x_{t,1}$. Therefore, maximizing the second term in (2) helps us **explore** the domain of prompts by promoting the selection of a prompt that is different from the previously selected prompts (including those in $\mathbf{X}_{t-1}$ and $x_{t,1}$). Here, $\nu$ is a parameter that controls the trade-off between exploration and exploitation.

In addition to being theoretically principled, another advantage of our prompt selection strategy is that it provides us with a natural method to choose the prompt to report as the best prompt. In POHF, we only have access to binary preference feedback between pairs of prompts and cannot observe numeric scores indicating the efficacy of different prompts. Therefore, it is non-trivial to choose which prompt to recommend as the best prompt. Interestingly, our strategy to select the first prompt provides a natural and principled way to choose the prompt to recommend. Specifically, after any iteration, we train the NN using the current history of preference observations, and choose the prompt (among all previously selected prompts) which maximizes the predicted score of the trained NN to report as the best prompt (line 6 of Algo. 1). This is in fact analogous to a common practice in Bayesian optimization, i.e., choosing the input (among all previously queried inputs) that maximizes the predicted function value (i.e., the Gaussian process posterior mean) to report as the best input (Nguyen et al., 2021).

## 3.3 Collecting User Preference Feedback

After the pair of prompts $x_{t,1}$ and $x_{t,2}$ are selected, we then separately pass them to the target black-box LLM to produce their corresponding responses. Next, these two responses are shown to the user, who then gives preference feedback $y_t = \mathbb{1}(x_{t,1}, x_{t,2})$ indicating which one of the two responses (generated from $x_{t,1}$ and $x_{t,2}$) is preferred. Then, the newly collected observation $(x_{t,1}, x_{t,2}, y_t)$ is added to the history of preference observations to yield $\mathcal{D}_t = \{(x_{s,1}, x_{s,2}, y_s)\}_{s=1,...,t}$, after which we use the updated history $\mathcal{D}_t$ to train our NN (Sec. 3.1) and proceed to the next iteration $t+1$.

In addition to the above-mentioned preference feedback, at the beginning of our APOHF, the user needs to provide some initial task description (Fig. 1), which our APOHF algorithm uses to generate the domain of prompts (Sec. 2). The initial task description may be in the form of some input-output exemplars for the task (we follow this in our experiments in Sec. 4.1), which our APOHF algorithm can use as input to a powerful LLM to produce the domain of prompts via *in-context learning* (Lin et al., 2024). As another example, the initial task description from the user may also be an initial

prompt for the task (we follow this in our experiments in Sec. 4.2), and our APOHF algorithm uses a powerful LLM (e.g., ChatGPT) to rephrase this initial prompt to produce the domain of prompts. This renders our APOHF algorithm highly flexible and versatile across a broad spectrum of real-world applications.

## 4 EXPERIMENTS

We test the performance of our APOHF using 3 sets of tasks: optimization of user instructions (Sec. 4.1), prompt optimization for text-to-image generative models (Sec. 4.2), and response optimization with human feedback (Sec. 4.3). To the best of our knowledge, our APOHF is the first algorithm that is designed to efficiently solve the problem of POHF. We compare our APOHF with 3 natural baseline methods which we adapt to POHF. (1) **Random Search** randomly selects a prompt in every iteration and hence ignores the preference feedback. (2) **Linear Dueling Bandits** (Bengs et al., 2022) uses a linear function to model the latent score function $u$ and adopts a strategy from Bengs et al. (2022) to select the pair of prompts (more details in App. C). After every iteration, the prompt predicted by the linear model to achieve the largest score is reported as the best prompt. (3) **Double Thompson Sampling (DoubleTS)** was recently applied to the problem of response optimization with human feedback by Dwaracherla et al. (2024) and was shown to be the best-performing method. We follow the implementation of DoubleTS from Dwaracherla et al. (2024): We choose the pair of prompts by independently running Thompson sampling (TS) twice, in which the reward/score uncertainty is modeled using Epistemic NNs (which consists of 10 individual MLPs). We also use TS to choose the prompt to report as the best prompt after every iteration. Note that DoubleTS incurs significantly more computational costs than our APOHF, mainly because DoubleTS needs to train 10 MLPs (in contrast to 1 MLP needed by our APOHF) in every iteration. For fair comparisons, we use 150 human preference feedback/iterations for each method in all experiments. All methods follow the framework of prompt optimization with human feedback (as shown in Fig. 1) and the only difference among different methods is how to select a pair of prompts. Note that **previous methods of prompt optimization cannot be applied to the scenarios we consider in this work because they require a scoring method**, therefore, we do not compare with these methods such as Zhou et al. (2023); Yang et al. (2024); Lin et al. (2024); Hu et al. (2024).

### 4.1 OPTIMIZATION OF USER INSTRUCTIONS

To begin with, we simulate real-world scenarios in which a user aims to find the optimal instruction for a task while only giving human preference feedback. We adopt 30 instruction induction tasks from Chen et al. (2023); Lin et al. (2024), which have been commonly used by previous works on instruction optimization for black-box LLMs (Chen et al., 2023; Hu et al., 2024; Lin et al., 2024). Here, we consider the scenario where the user provides a small number of input-output exemplars as the initial task description (Fig. 1), and we use these exemplars to generate the domain of prompts for our APOHF via in-context learning (Sec. 3.3). Specifically, to generate each prompt/instruction in the domain, we randomly sample 5 exemplars from the dataset of 100 exemplars (which are separate from the validation set), and ask ChatGPT to generate the instruction that best describes the input-output relationship of these 5 exemplars via in-context learning. We provide the ChatGPT template used to generate the instructions in Example 1 (App. A.3). At each iteration in Fig. 1, we use our APOHF to select a pair of instructions/prompts to obtain the human preference feedback.

**Simulation of human feedback.** Since human preference feedback is not available in this experiment, we propose to simulate the human preference feedback by using a validation dataset of input-output exemplars. Specifically, after selecting a pair of instructions/prompts $x_{t,1}$ and $x_{t,2}$, we use the validation dataset for this task to calculate the validation accuracy achieved by both instructions, which we adopt as their ground-truth latent score values: $u(x_{t,1})$ and $u(x_{t,2})$. Then, we calculate the preference probability $\mathbb{P}(x_1 \succ x_2) = \sigma(u(x_1) - u(x_2))$, and use it as the probability in a Bernoulli distribution to sample the binary preference observation $y_t = \mathbb{1}(x_{t,1} \succ x_{t,2})$. This also naturally allows us to report the validation accuracy achieved by an instruction $x$ as its corresponding latent score value $u(x)$, which we plot in our results (Fig. 2). Of note, unlike the previous works (Chen et al., 2023; Hu et al., 2024; Lin et al., 2024), the validation dataset for each task is not used by our algorithm; instead, it is only used to simulate the human preference feedback.

Fig. 2 displays the performances of different methods averaged over 30 tasks. After each iteration, every method reports a prompt as the best prompt, and its corresponding latent score (i.e., validation accuracy in this case) is plotted in Fig. 2. The figure shows that our APOHF algorithm consistently and significantly outperforms the other methods. This is because our APOHF has a better selection strategy (as described in Sec. 3.2) for selecting the pair of prompts in each iteration. We also demonstrate the progression of the best instruction discovered by our APOHF in Table 1, which further illustrates the capability of our APOHF to efficiently find good instructions using only preference feedback.

## 4.2 Prompt Optimization for Text-to-Image Generative Models

Modern text-to-image generative models, such as DALLE-3 (Betker et al., 2023), have shown remarkable capabilities in generating visually appealing images (Chen et al., 2024a; Rombach et al., 2022; Song et al., 2020a). These models take a text prompt as input and generate a corresponding image. When a user adopts DALLE-3 to generate an image, they may need to manually try a number of different prompts in order to obtain a desirable image. Interestingly, in such applications, our APOHF algorithm can also be adopted to efficiently find the best prompt for a user. Specifically, in every iteration, we can use our APOHF algorithm to select a pair of text prompts and generate two corresponding images using DALLE-3, and then ask the user for preference feedback between the two images. We simulate such scenarios using the experiments in this section.

To begin with, we adopt an initial prompt (as the initial task description in Fig. 1) that describes a complex scene using several sentences (see App. 3 for more details), and rephrase the initial prompt to produce a large number of text prompts (more details in App. A). These prompts are used as the domain of prompts for our APOHF. In each iteration, our APOHF select a pair of prompts from the domain to generate two images and obtain the user feedback. In this case, the goal of our APOHF is to efficiently find a prompt from the domain of prompts to produce an image that is most preferred by a user, while only requiring a small number of user preference feedback instances.

**Simulation of human feedback.** To simulate the user feedback, we select one of the prompts from the domain as the ground-truth prompt. Our implicit assumption is that *the image generated by this ground-truth prompt is the image which is most desirable by the user*. Therefore, for every candidate prompt $x$ in the domain, we measure the similarity of its generated image with the image generated by the ground-truth prompt and use the similarity as the latent score $u(x)$ of this prompt. As a result, for every pair of selected prompts $x_{t,1}$ and $x_{t,2}$, we can calculate their preference probability using the BTL model: $\mathbb{P}(x_1 \succ x_2) = \sigma(u(x_1) - u(x_2))$, and then sample a binary preference observation $y_t$ from a Bernoulli distribution with probability $\mathbb{P}(x_1 \succ x_2)$.

We repeat the experiment for 4 different scenes and report the scores of different methods in Fig. 3. The results show that our APOHF consistently outperforms the other baselines across different scenes. That is, our APOHF is able to efficiently discover a prompt to generate an image that satisfies the user's preferences. We also demonstrate in Fig. 4 the evolution of the images generated by the best prompts discovered by our APOHF across different iterations. The results suggest that as more user feedback is collected, *our APOHF can efficiently produce images which better align with the image the user has in mind.* Note that here we intend for the generated images to match the high-level semantic information of the ground-truth image rather than the image details, which are usually uncontrollable due to the inherent randomness in image generation. This experiment showcases the considerable potential of our APOHF beyond text-generation tasks, suggesting its applicability to a wide range of multi-modal tasks where using human feedback is preferable.

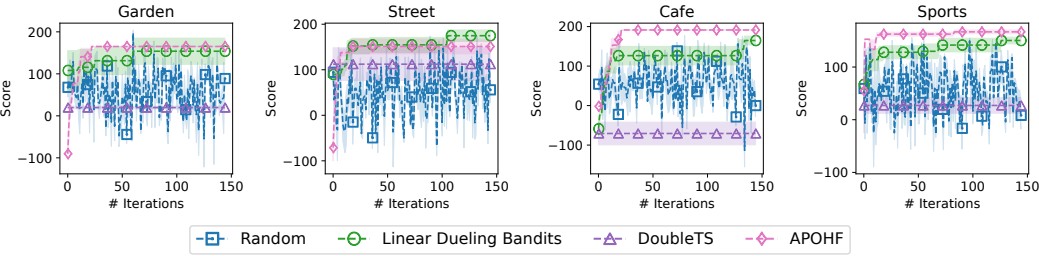

Figure 3: Performances in prompt optimization for image generation in Sec. 4.2 (4 different scenes).

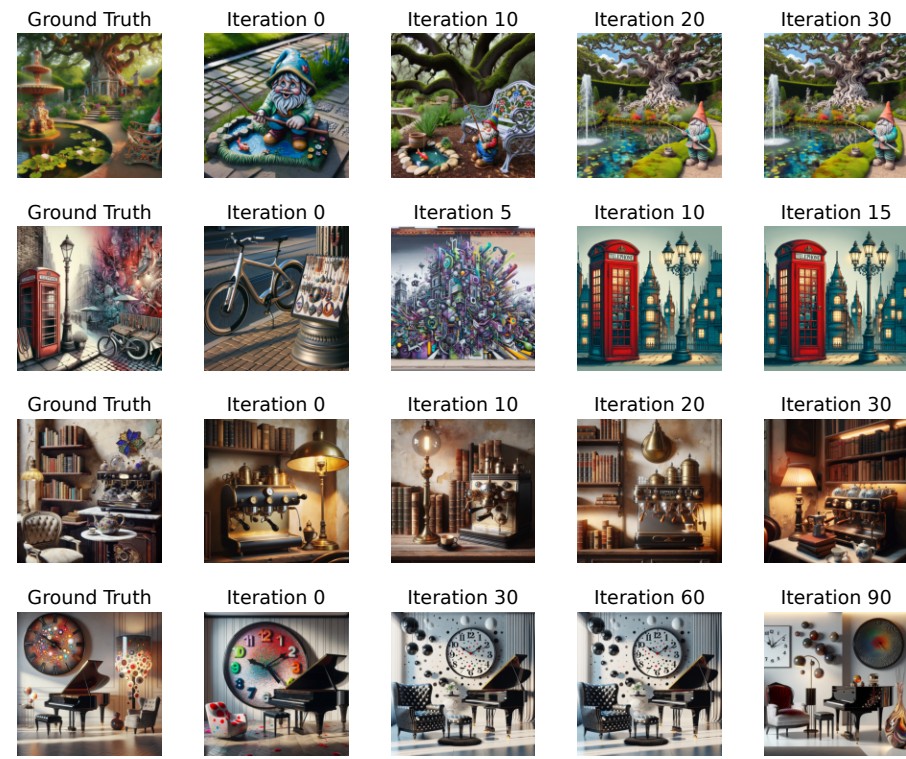

Figure 4: Images generated by the best prompt discovered by our APOHF across different iterations.

### 4.3 Response Optimization with Human Feedback

In addition to adapting the response of an LLM by optimizing the prompt (i.e., by solving POHF), our APOHF algorithm can also be used to further refine the response from the LLM by tackling the problem of *response optimization with human feedback* (Sec. 1). Specifically, given a prompt from a user, we can let the LLM generate a large pool of responses and then try to choose the best response from the pool. Similar to POHF, instead of requesting the user for a numeric score, it is much easier to ask the user for preference feedback between a pair of responses (Sec. 1). This problem setting has also been adopted by the recent work of Dwaracherla et al. (2024).

This problem can be tackled by a *contextual* variant of our APOHF. That is, every prompt $p$ can be seen as a *context*, and the pool of responses $r$'s generated from this prompt can be considered the domain of *actions*. Here, we need to make an important modification to our APOHF. That is, in iteration $t$ after receiving the prompt $p_t$, every input $x$ in the domain is now the embedding of the concatenation of the prompt $p_t$ and one of the LLM-generated responses $r$, which we denote as $x = [p_t, r]$. As an implication, the domain $\mathcal{X}_t$ from which we choose a pair of inputs changes in every iteration (as a result of the changing prompt $p_t$). However, the strategy for selecting the pair of inputs remains the same (Sec. 3.2), except that the fixed domain $\mathcal{X}$ is now replaced by the changing domain $\mathcal{X}_t$.

**Simulation of human feedback.** To simulate the user preferences between different responses, we adopt the same approach as Dwaracherla et al. (2024). That is, we use a reward model which is pre-trained using the Anthropic Helpfulness and Harmlessness datasets (Bai et al., 2022). Then, given a user prompt $p_t$, for every LLM-generated response $r$, we use the output from the pre-trained reward model as the latent score value $u([p_t, r])$ for this prompt-response pair. Then, for every pair of selected responses $r_{t,1}$ and $r_{t,2}$ by our APOHF, we can calculate the preference probability following the BTL model $\mathbb{P}\{r_{t,1} \succ r_{t,2}\} = \sigma(u([p_t, r_{t,1}]) - u([p_t, r_{t,2}]))$ and then use it to sample a binary preference observation $y_t$.

Table 1: The best instructions selected by our APOHF in different iterations (Sec. 4.1). Full table can be found in Table 6.

| Task | Iter | Instruction | Score |
|---|---|---|---|
| antonyms | 0 | add the prefix "un-" to the given words to form their opposites. | 0.45 |
| | 5 | remove the "un-" prefix from each word. | 0.45 |
| | 10 | provide the opposite of the given words. | 0.70 |
| larger animal | 0 | choose the second animal in each pair, so the output is the second animal in each pair. | 0.30 |
| | 60 | choose the animal that is not a type of fish, and if both animals are not fish, then choose the first animal. | 0.55 |
| | 120 | choose the animal that is larger in size, so the output would be: wildebeest shark elk pit bull manatee | 1.00 |
| sentiment | 0 | provide negative responses to the given inputs. | 0.00 |
| | 60 | provide an output based on the given input. | 0.00 |
| | 120 | provide the sentiment (positive/negative) of the given inputs. | 0.90 |
| word sorting | 0 | "Please alphabetize the following list of words." | 0.40 |
| | 30 | rearrange the words in the list in alphabetical order and the output provided is the rearranged list of words. | 0.75 |
| | 60 | rearrange the words in the list in alphabetical order and output the sorted list. | 0.85 |

Table 2: The best responses selected by our APOHF across iterations (Sec. 4.3). The prompt is "Human: What is there to do in Atlantic City?". We omit some words here (replaced by "..") for shorter display. Full table in App. 5.

| Iter | Response | Score |
|---|---|---|
| 0 | There are many activities to do in Atlantic City! You can visit the famous Atlantic City Boardwalk, try your luck at the casinos, enjoy the beach and water activities, shop at the outlets and specialty stores, dine at top-rated restaurants, enjoy live entertainment and nightlife, visit museums and historical sites, and attend events and festivals throughout the year. | 81.96 |
| 10 | There are plenty of things to do in Atlantic City! You can visit the famous boardwalk, try your luck at the casinos, enjoy a show or concert at one of the many entertainment venues, dine at a variety of restaurants, shop at the outlets and specialty stores, relax on the beach, or take a stroll through the historic district. There's something for everyone in Atlantic City! | 91.46 |
| 20 | There are many attractions and activities in Atlantic City, including: 1. Casinos: Atlantic City is known for its many casinos, where you can try your luck at slots, poker, blackjack, and more. 2. Boardwalk: ... 3. Beach: .. 4. Steel Pier: ... 5. Shows and concerts: ... 6. Nightlife: ... 7. Dining: ... 8. Shopping: ... | 180.14 |

The results are shown in Fig. 5, in which our APOHF significantly outperforms the other methods, including DoubleTS, which is found to be the best-performing method in Dwaracherla et al. (2024). We also show an example of how the response optimized by our APOHF is improved across iterations in Table 2. The response discovered by our APOHF after only 20 iterations is both well organized (via a numbered list) and detailed, which aligns well with human preferences. This demonstrates the ability of our APOHF to *further refine the response of an LLM to make it more preferable for human users*, while only requiring human preference feedback.

# 5 ABLATION STUDY

## 5.1 EFFECTIVENESS OF OUR PROMPT SELECTION STRATEGY

Here, we further verify the effectiveness of our theoretically principled prompt selection strategy. We replace the strategy of our APOHF to select a pair of prompts by uniform random selection while keeping all other components of our APOHF fixed. That is, after every iteration, we still train the NN using the current history of observations as described in Sec. 3.1, and report the prompt maximizing the prediction of the NN as the best prompt. The results (Fig. 6) show that randomly selecting the pair of prompts significantly degrades the performance of our APOHF, further validating the effectiveness of our prompt selection strategy (Sec. 3.2).

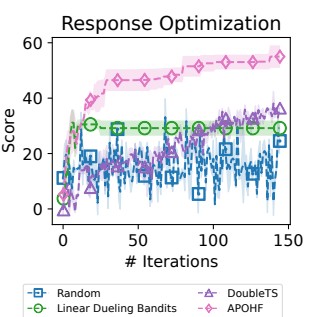

Figure 5: Scores of different methods for response optimization (Sec. 4.3).

## 5.2 IMPACT OF THE EXPLORATION PARAMETER

Here, we examine the impact of the exploration parameter $\nu$ on our APOHF algorithm. The results (Fig. 7) show that setting $\nu = 0$ (i.e., not performing any exploration) degrades the performance of our APOHF. This is because it limits the ability of our APOHF to sufficiently explore the space of possible prompts. On the other hand, using a large value of $\nu = 10$ does not significantly affect the performance of APOHF. This is because although a large $\nu$ may result in excessive exploration when selecting the second prompt, the value of $\nu$ does not alter our strategy to choose the first prompt. Therefore, a large exploration parameter $\nu$ does not significantly diminish the ability of our APOHF to exploit the prediction of the NN.

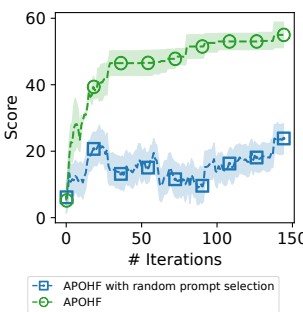 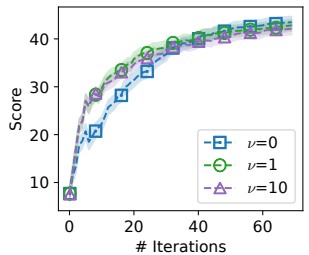

Figure 6: Comparison of our arm selection strategy with random selection.

Figure 7: Comparison of the performance of our `APOHF` algorithm with different values of $\nu$ (i.e., the exploration parameter).

### 5.3 Impact of the Level of Noise in Preference Feedback

Here, we study the impact of the level of noise in preference feedback on the performance of different algorithms. We alter the level of noise in preference feedback by adjusting the scale of the latent score function $u$. A smaller scale of the scores results in noisier preference observations and hence leads to a more difficult optimization problem. This is because according to the BTL model $\mathbb{P}(x_1 \succ x_2) = \sigma(u(x_1) - u(x_2))$, a smaller scale of $u(\cdot)$ generally makes the preference probability closer to $0.5$. This renders the resulting binary observation $y_t = \mathbb{1}(x_{t,1} \succ x_{t,2})$ more similar to a purely random sample (with a probability of $0.5$) and hence noisier. The results (Fig. 8) verify that the smaller the noise, the more pronounced the advantage of our `APOHF`. Meanwhile, as the noise level becomes too large, the problem becomes excessively difficult for all methods, and eventually, all algorithms achieve similar performances.

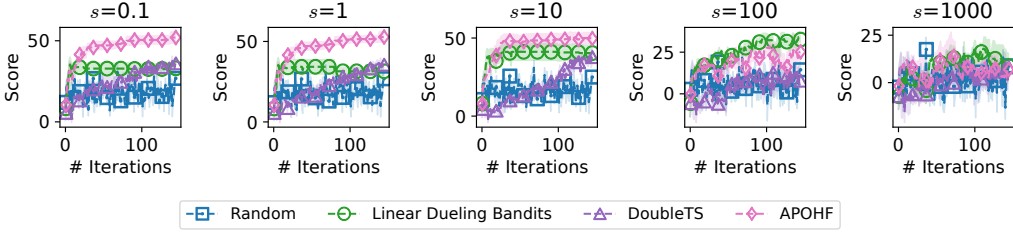

Figure 8: Comparison of the performances of different algorithm under different levels of noise in human feedback. Here $s$ controls the level of noise, such that a larger $s$ results in a higher noise level.

## 6 Related Work

Prompt optimization, also referred to as *instruction optimization*, has been gaining popularity thanks to its ability to improve the performance of LLMs without parameter fine-tuning. Earlier works aimed to optimize the prompt for white-box LLMs, such as AutoPrompt (Shin et al., 2020), FluentPrompt (Shi et al., 2023), as well as other works based on soft prompt (Lester et al., 2021; Li & Liang, 2021; Zhong et al., 2021). Recently, more focus has been shifted to optimizing the prompt for black-box LLMs. Among them, BBT (Sun et al., 2022b), BBTv2 (Sun et al., 2022a) and Clip-Tuning (Chai et al., 2022) require access to the input embedding and output logits of the black-box LLM. Other recent works have removed this restriction. For example, GRIPS (Prasad et al., 2023) and APO (Pryzant et al., 2023) used edit-based operations to select candidate prompts for prompt optimization. Other works have adopted evolutionary algorithms (e.g., EvoPrompt (Guo et al., 2024) and Promptbreeder (Fernando et al., 2023)), reinforcement learning (e.g., BDPL (Diao et al., 2023) and PRewrite (Kong et al., 2024)), and planning-based methods (e.g., PromptAgent (Wang et al., 2023b)) to achieve prompt optimization for black-box LLMs. The work of Zhou et al. (2023) proposed APE, which generates candidate instructions using an LLM and selects those high-scoring candidates for further refinement. The OPRO algorithm (Yang et al., 2024) was developed to use an LLM to solve generic black-box optimization problems and was applied to the problem of prompt optimization. The work of Mañas et al. (2024) introduced OPT2I, which uses an LLM to sequentially revise the prompt

for text-to-image generative models, in order to maximize a score measuring the consistency of the generated image with the given prompt.

Some recent works have tackled prompt optimization for black-box LLMs by converting it to a continuous optimization problem. InstructZero (Chen et al., 2023) adopted a separate white-box LLM to convert prompt optimization to optimizing the soft prompt and used Bayesian optimization to solve the resulting continuous optimization problem. INSTINCT (Lin et al., 2024) used neural bandits to sequentially select the instructions to query and leveraged the strong expressive power of neural networks to achieve better function modeling and hence better prompt optimization. ZOPO (Hu et al., 2024) adopted zeroth-order optimization (ZOO) while estimating the gradient based on a neural network, and further improved the performances of InstructZero and INSTINCT. In addition, Shi et al. (2024) demonstrated the potential of drawing inspirations from best arm identification for prompt optimization, and Chen et al. (2024b) used neural bandits for personalized content generation using white-box LLMs. Importantly, to the best of our knowledge, *these previous works are not able to tackle the problem of POHF considered in our work*, because they require a numeric score to evaluate the efficacy of each prompt.

RLHF has become the most widely used method for aligning the responses of LLMs with human values (Dubois et al., 2024; Ouyang et al., 2022; Ziegler et al., 2019). More comprehensive discussions on RLHF can be found in recent surveys (Casper et al., 2023; Chaudhari et al., 2024). More recently, some methods have been developed to sidestep the need for RL and directly use a preference dataset for alignment, including direct preference optimization (DPO) (Rafailov et al., 2024), SLiC (Zhao et al., 2023), as well as other extensions (Amini et al., 2024; Azar et al., 2024; Gou & Nguyen, 2024; Liu et al., 2024; Morimura et al., 2024; Tang et al., 2024; Wang et al., 2023a). The recent work of Dwaracherla et al. (2024) has shown the potential of efficient exploration methods to improve the response of LLMs with human preference feedback.

## 7 CONCLUSION AND LIMITATIONS

We have introduced the problem of POHF, in which our goal is to optimize the prompt for black-box LLMs while using only human preference feedback. To address POHF, we have proposed the APOHF algorithm, which uses a neural network trained using preference feedback to model the latent score function, and chooses the pair of prompts to query based on a principled strategy inspired by dueling bandits. By using various tasks, including user instruction optimization, prompt optimization for text-to-image generative models, and response optimization with human feedback, we empirically validate that our APOHF is able to find a good prompt for a task using a small number of human feedback instances. A potential limitation of our APOHF is that it currently does not accommodate the scenario where more than 2 prompts are selected in every iteration, and the user provides feedback regarding the ranking of the responses from these prompts. We plan to tackle this in future work by developing novel and theoretically principled strategies to choose more than 2 prompts to query.

## 8 REPRODUCIBILITY STATEMENT

We have provided the source code for our experiments in the supplemental materials to ensure reproducibility. We have also provided the details of datasets, computational resources, and hyper-parameters for our experiments in Appendix A.

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

# A    ADDTIONAL DETAILS FOR EXPERIMENTS

## A.1    LICENSE FOR DATASETS

(1) Instruction induction dataset (Chen et al., 2023; Lin et al., 2024) for optimizing the user instruction: MIT License; (2) Anthropic Helpfulness and Harmlessness datasets (Bai et al., 2022) for response optimization: MIT License.

## A.2    COMPUTATIONAL RESOURCES

All the experiments are run on a server with AMD EPYC 7763 64-Core Processor, 1008GB RAM, and 8 NVIDIA L40 GPUs.

## A.3    ADDITIONAL DETAILS ON EXPERIMENTAL SETTINGS

**Hyper-parameters.**    We use an MLP with 2 hidden layers as the NN for the latent score prediction. Each hidden layer has a width of 32. At each iteration of our APOHF we re-initialize the NN and train the NN using all available human feedback data for 1000 epochs with Adam optimizer and a learning rate of 0.001. We run all algorithms for 150 iterations. We normalize the score distributions for all applications to $\mathcal{N}(0, 100)$ such that the simulated feedback obtained by the BTL model will not be too noisy. We use the hyper-parameters of $\nu = 1$ and $\lambda = 0.1$ for our APOHF and Linear Dueling Bandits. For the prompt optimization for text-to-image generative models, we use a larger $\nu = 10$ for both algorithms for better exploration. All the experiments are run at least 2 times to obtain the error bars and the average performances. For ChatGPT queries used in all experiments, we use the specific version of "gpt-3.5-turbo-1106" API provided by OpenAI.

**User instruction optimization.**    We generate a prompt domain with 200 prompts/instructions. The validation dataset has a size of 20. The exemplar dataset provided by the user has a size of 100. The validation accuracy for a prompt/instruction is evaluated by using the validation dataset and querying ChatGPT, which is the same as previous works (Chen et al., 2023; Lin et al., 2024). We use MPNet (Song et al., 2020b) to obtain the representations of the prompts to be the inputs to our NN for the latent score prediction.

**Prompt optimization for text-to-image generative models.**    We generate a prompt domain with 200 prompts. Specifically, we use the template in Example 2 to rephrase the initial prompt for each scene in Table 3 to obtain the ground-truth prompt. We use the template in Example 2 to rephrase the initial prompt again to obtain 10 different prompts as good candidates in the prompt domain. This is to make sure that the domain contains some prompts that are very close to the ground-truth prompt. For the generation of the other 190 prompts in the domain, we first select a subset of sentences from the initial prompt. Specifically, each sentence in the initial prompt is selected with a probability of 0.3 independently. This is to simulate real-life scenarios where the prompts provided by the users may only contain a fraction of the information needed to generate the ground-truth or ideal images. We combine the selected subset of sentences to form a new prompt and use the template in Example 3 to rephrase it to obtain a new element in the prompt domain. We repeat the above procedures to obtain the other 190 prompts. We use the DALLE-3 model with the generation quality as "standard" and the generation size as "$1024 \times 1024$". We use CLIP (Radford et al., 2021) to obtain the representations of the ground-truth image and the generated images. We use the cosine similarity function to calculate the similarity score between the representations of the ground-truth image and the generated image as the quality measure for the corresponding generated image. We use vision transformer (Dosovitskiy et al., 2020) to obtain the representations of the generated images to be the inputs to the NN for the latent score prediction. The reason for using a different representation model for the latent score prediction is to simulate real-life scenarios in which we do not have prior knowledge about the ground-truth score function.

**Response optimization with human feedback.**    We randomly select 10 questions from the test dataset of the Anthropic Helpfulness dataset as the prompts. For each prompt, we generate 50 responses from ChatGPT. We set the temperature parameter of ChatGPT to be 1.0 so that the generated responses are different from each other. We use a fine-tuned GPT-2 model (Radford

Table 3: Initial prompts for generating images with different scenes.

| Scene | Prompt |
|-------|--------|
| Garden | In a vibrant garden, a grand marble fountain gushes clear water, dazzling in the sunlight. Nearby, a centuries-old oak tree stands with sprawling, gnarled branches. A vintage wrought iron bench with floral patterns offers a quaint seat. Beside the path, a whimsical, brightly painted gnome statue holds a fishing rod towards a small pond. In the pond, lily pads float with blooming white lilies. |
| Street | On a lively city street, a striking vintage red telephone booth pops against the muted city colors. Nearby, a vibrant graffiti mural adds color to a plain brick wall, featuring an abstract mix of urban elements. A futuristic bicycle with a shiny, aerodynamic silver frame is locked to a lamppost. A small vendor's stall on the sidewalk displays handmade, colorful beaded jewelry, glistening in the afternoon sun. In the background, an ornate old-fashioned street lamp emits a warm glow as dusk approaches. |
| Cafe | In a quaint cafe corner, a vintage espresso machine with polished brass fixtures and a matte black body gleams under an antique lamp. A rustic wooden bookshelf, brimming with well-worn books, stands against a distressed cream wall. A marble table at the room's center holds a delicate porcelain teapot with intricate blue flowers, from which steam gently rises. Beside the table, a colorful glass mosaic cat sculpture perches on a mismatched velvet chair, casting playful reflections around. |
| Sports | A sleek grand piano with a glossy black surface speckled with white spots stands at the room's center. On the wall, a colorful clock features a face marked by vibrant, multicolored spots for each hour. Beside it, a tall floor lamp sports a leopard-spot patterned lampshade in black and gold. A plush armchair in the corner showcases bold red polka dots on a white background. On a nearby table, a delicate glass vase captivates with swirling, iridescent spots that shimmer in the light. |

et al., 2019) to obtain the ground-truth scores for prompt-response pairs. Specifically, the GPT-2 is fine-tuned on Anthropic Helpfulness dataset (Bai et al., 2022) to determine the helpfulness of the response w.r.t. a prompt by outputting a score. For each response, we concatenate its corresponding prompt as the prefix and input it to the fine-tuned GPT-2 model to obtain the ground-truth score. We use MPNet to obtain the representations of the prompt-response pairs to be the inputs to our NN for the latent score prediction. For each iteration of the algorithm, a prompt is selected in a round-robin fashion with a fixed order. This is for the purpose of result visualization and fair comparison since different algorithms select responses for the same prompt in each corresponding iteration, and their performances are evaluated based on the same domain in this way. The score for each algorithm in an iteration in Fig. 5 is calculated by using the trained latent score prediction model in this iteration from each algorithm to select the best responses for each of the 10 prompts and evaluating these 10 prompt-response pairs with the ground-truth score function to obtain an average score.

---

**Example Query 1: Instruction Induction Template**

Input: [INPUT]
Output: [OUTPUT]

<More exemplars...>

Input: [INPUT]
Output: [OUTPUT]

The instruction was to:

---

Table 4: The best responses selected by our APOHF in different iterations. The prompt is "Human: Should you buy life insurance? Assistant: I'm sure life insurance is useful for a lot of people. I can also tell you how you can get insurance, depending on your income and needs. Human: Is term or whole life insurance better?"

| Iter | Response | Score |
|---|---|---|
| 0 | It really depends on your individual circumstances and financial goals. Term life insurance provides coverage for a specific period of time, while whole life insurance provides coverage for your entire life and also has an investment component. It's important to carefully consider your financial situation and consult with a financial advisor before making a decision. | 45.54 |
| 10 | Term life insurance is typically more affordable and provides coverage for a specific period of time, while whole life insurance provides coverage for your entire life and also includes a cash value component. The best option for you will depend on your individual needs and financial situation. It's important to carefully consider your options and consult with a financial advisor or insurance expert. | 100.96 |
| 20 | Term life insurance is typically more affordable and provides coverage for a specific period of time, while whole life insurance provides coverage for your entire life and also includes a cash value component. The best option for you will depend on your individual needs and financial situation. It's important to carefully consider your options and consult with a financial advisor or insurance expert. | 100.96 |

---

**Example Query 2: Image Generation Instruction Rephrasing Template 1**

Rephrase the following description: [Initial instruction]
The rephrased description is:

---

**Example Query 3: Image Generation Instruction Rephrasing Template 2**

Modify the following description by rephrasing and changing some information: [Initial instruction]
The modified description is:

---

## B  ADDTIONAL EXPERIMENTAL RESULTS

Table 4 and Table 5 show the best responses selected by our APOHF in different iterations. An observation is that both the response quality and the score increase when more iterations are performed. Specifically, for Table 4, the response in iterations 10 and 20 include more information on the affordability of different kinds of insurance compared to iteration 0. For Table 5, as more iterations of our APOHF are performed, the response provides more details on the things to do in Atlantic City. In iteration 20, the response even includes bullet points for the recommended activities for the users and hence is the most useful response.

Table 6 shows the best instructions selected by our APOHF for different tasks. As more iterations are performed, the instructions become more relevant to the tasks and hence have higher scores.

## C  THEORETICAL JUSTIFICATIONS FOR OUR APOHF ALGORITHM

Here, we provide theoretical justifications for the prompt selection strategy of our APOHF algorithm, to show that our APOHF algorithm is theoretically principled. Our goal here is to provide high-level intuitions rather than giving a complete theoretical analysis.

Recall that as we have introduced in Sec. 3.2, our APOHF algorithm selects the first prompt greedily by maximizing the predicted score (from the trained NN), and chooses the second prompt by maximizing upper confidence bound, which is a weighted combination of the score prediction and an uncertainty

Table 5: The best responses selected by our `APOHF` in different iterations. The prompt is "Human: What is there to do in Atlantic City?".

| Iter | Response | Score |
|---|---|---|
| 0 | There are many activities to do in Atlantic City! You can visit the famous Atlantic City Boardwalk, try your luck at the casinos, enjoy the beach and water activities, shop at the outlets and specialty stores, dine at top-rated restaurants, enjoy live entertainment and nightlife, visit museums and historical sites, and attend events and festivals throughout the year. | 81.96 |
| 10 | There are plenty of things to do in Atlantic City! You can visit the famous boardwalk, try your luck at the casinos, enjoy a show or concert at one of the many entertainment venues, dine at a variety of restaurants, shop at the outlets and specialty stores, relax on the beach, or take a stroll through the historic district. There's something for everyone in Atlantic City! | 91.46 |
| 20 | There are many attractions and activities in Atlantic City, including:
1. Casinos: Atlantic City is known for its many casinos, where you can try your luck at slots, poker, blackjack, and more.
2. Boardwalk: Take a stroll on the iconic Atlantic City Boardwalk, lined with shops, restaurants, and amusement attractions.
3. Beach: Enjoy a day of sun and sand at the Atlantic City beach, a popular spot for swimming, sunbathing, and water sports.
4. Steel Pier: Visit this historic amusement park featuring rides, games, and entertainment for the whole family.
5. Shows and concerts: Catch a live performance or concert at one of the many entertainment venues in Atlantic City, including the Hard Rock Hotel & Casino and the Borgata Hotel Casino & Spa.
6. Nightlife: Experience the vibrant nightlife of Atlantic City, with numerous bars, nightclubs, and lounges offering live music, DJs, and dancing.
7. Dining: Indulge in a variety of dining options, from upscale restaurants to casual eateries serving fresh seafood, steaks, and international cuisine.
8. Shopping: Explore the shops and boutiques in Atlantic City, offering everything from designer fashions to unique souvenirs. | 180.14 |

term (2). This strategy is inspired by previous works on linear dueling bandits (Bengs et al., 2022; Saha, 2021).

Here, we adopt the simplifying assumption that the utility/score function $u$ is a linear function: $u(x) = \theta^\top x, \forall \mathcal{X}$ with unknown parameter $\theta$. With this assumption, our prompt selection strategy can be seen as a modified version of the algorithm from (Bengs et al., 2022). Therefore, we follow the notations from (Bengs et al., 2022) and present below the most important modifications to the theoretical analysis of (Bengs et al., 2022). We use $z_{1,2}$ to denote the difference between (the features vectors of) a pair of prompts: $z_{1,2} = x_1 - x_2$ and use $z_{t,1,2} = x_{t,1} - x_{t,2}$ to denote the difference between the pair of selected prompts in iteration $t$. The matrix $M_t = \sum_{s=1}^{t} z_{t,1,2}^\top z_{t,1,2}$ intuitively characterizes the information collected up to iteration $t$.

With these notations, $\hat{\theta}^\top z$ represents the *predicted reward difference* between a pair of prompts $x_1$ and $x_2$, which in our case, corresponds to $h(x_1; \theta_t) - h(x_2; \theta_t)$. Then, $\theta^\top z$ represents the ground-truth reward difference. Following the standard practice of the analysis of bandit algorithms (Bengs et al., 2022), we assume that the validity of the confidence bound providing a theoretical guarantee on the quality of reward difference estimation: $|\theta^\top z - \hat{\theta}^\top z| \leq \nu \|z\|_{M_t^{-1}}$. With these, the *regret* incurred in iteration $t$ can be analyzed as:

$$2r_t = u(x^*) - u(x_{t,1}) + u(x^*) - u(x_{t,2})$$
$$\stackrel{(a)}{=} \theta^\top(x^* - x_{t,1}) + \theta^\top(x^* - x_{t,2})$$
$$\stackrel{(b)}{=} \theta^\top z_{t,1}^* + \theta^\top z_{t,2}^*$$
$$= (\theta - \hat{\theta}_t)^\top z_{t,1}^* + \hat{\theta}_t^\top z_{t,1}^* + (\theta - \hat{\theta}_t)^\top z_{t,2}^* + \hat{\theta}_t^\top z_{t,2}^*$$

Table 6: The best instructions selected by our APOHF in different iterations.

| Task | Iter | Instruction | Score |
|------|------|-------------|-------|
| antonyms | 0 | add the prefix "un-" to the given words to form their opposites. | 0.45 |
| | 5 | remove the "un-" prefix from each word. | 0.45 |
| | 10 | provide the opposite of the given words. | 0.70 |
| informal to formal | 0 | rephrase the given sentences, so I have provided the rephrased versions of the input sentences as output. If this is not what you were looking for, please provide more specific instructions. | 0.39 |
| | 5 | rephrase the given sentences using formal language. | 0.44 |
| | 10 | rephrase each input sentence using a more formal or polite tone. | 0.47 |
| larger animal | 0 | choose the second animal in each pair, so the output is the second animal in each pair. | 0.30 |
| | 60 | choose the animal that is not a type of fish, and if both animals are not fish, then choose the first animal. | 0.55 |
| | 120 | choose the animal that is larger in size, so the output would be: wildebeest shark elk pit bull manatee | 1.00 |
| orthography starts with | 0 | identify the word in the sentence that is in Russian, and for the first three sentences, the word "Russian" was correctly identified. However, for the last two sentences, there were no words in Russian, so the output should have been "N/A" or "none." | 0.00 |
| | 20 | identify the adjective in each sentence. | 0.15 |
| | 40 | provide the word that starts with the given letter. | 0.80 |
| rhymes | 0 | change the first letter of the word to "inv" and then add the rest of the word. | 0.00 |
| | 4 | find a word that is an anagram of the given word. | 0.00 |
| | 8 | change the word to a new word that rhymes with the original word. | 0.40 |
| second word letter | 0 | "Provide the index of the first occurrence of the letter 'a' in each word." | 0.00 |
| | 2 | "Provide the index of the first occurrence of the letter 'a' in each word." | 0.00 |
| | 4 | "Output the second letter of each word." | 1.00 |
| sentiment | 0 | provide negative responses to the given inputs. | 0.00 |
| | 60 | provide an output based on the given input. | 0.00 |
| | 120 | provide the sentiment (positive/negative) of the given inputs. | 0.90 |
| taxonomy animal | 0 | rearrange the words in alphabetical order, so the output for each input would be the words listed in alphabetical order. | 0.00 |
| | 30 | rearrange the words in alphabetical order, so the output lists the words in alphabetical order. | 0.00 |
| | 60 | "Output the animals from the given list." | 0.95 |
| word sorting | 0 | "Please alphabetize the following list of words." | 0.40 |
| | 30 | rearrange the words in the list in alphabetical order and the output provided is the rearranged list of words. | 0.75 |
| | 60 | rearrange the words in the list in alphabetical order and output the sorted list. | 0.85 |

$$\overset{(c)}{\leq} \hat{\theta}_t^\top z_{t,1}^* + \nu \left\| z_{t,1}^* \right\|_{M_t^{-1}} + \hat{\theta}_t^\top z_{t,2}^* + \nu \left\| z_{t,2}^* \right\|_{M_t^{-1}}$$

$$\overset{(d)}{\leq} 2\hat{\theta}_t^\top (x^* - x_{t,1}) + 2\nu \left\| x^* - x_{t,1} \right\|_{M_t^{-1}} + \hat{\theta}_t^\top z_{t,1,2} + \nu \left\| z_{t,1,2} \right\|_{M_t^{-1}}$$

$$\overset{(e)}{\leq} 2\hat{\theta}_t^\top (x_{t,2} - x_{t,1}) + 2\nu \left\| x_{t,2} - x_{t,1} \right\|_{M_t^{-1}} + \hat{\theta}_t^\top (x_{t,1} - x_{t,2}) + \nu \left\| x_{t,1} - x_{t,2} \right\|_{M_t^{-1}}$$

$$\leq \hat{\theta}_t^\top (x_{t,2} - x_{t,1}) + 3\nu \left\| z_{t,1,2} \right\|_{M_t^{-1}}$$

$$\overset{(f)}{\leq} 3\nu \left\| z_{t,1,2} \right\|_{M_t^{-1}}. \tag{3}$$

Step $(a)$ follows because here we have assumed that the score function $u$ is a linear function; in step $(b)$, we have defined $z_{t,1}^* = x^* - x_{t,1}$ and $z_{t,2}^* = x^* - x_{t,2}$; step $(c)$ follows because we have assumed the validity of the confidence bound as described above; step $(d)$ follows simply because $z_{t,2}^* = x^* - x_{t,2} = x^* - x_{t,1} + x_{t,1} - x_{t,2} = z_{t,1}^* + z_{t,1,2}$ (we have also made use of the triangle inequality).

**Selection of the Second Prompt.** Step $(e)$ follows from the way the second prompt is selected: $x_{t,2} = \arg\max_{x \in \mathcal{X}} \hat{\theta}_t^\top x + \nu \left\| x - x_{t,1} \right\|_{M_t^{-1}}$. This, importantly, is analogous to the way in which our APOHF algorithm selects the second prompt using Eq. (2). Note that we have replaced the linear score prediction $\hat{\theta}_t^\top x$ by the prediction from our NN: $h(x; \theta_t)$. We have also used the gradient $\nabla h(x; \theta_t)$ to replace the original feature vector $x$, which is justified by the theory of the neural tangent kernel (NTK), which has shown that $\nabla h(x; \theta_t)$ can be used to approximate the random Fourier features for the NTK (Jacot et al., 2018). Also note that compared to the theory of NTK, we have designed our APOHF algorithm to be more practical following the common practice of neural bandits (Zhang et al., 2021; Zhou et al., 2020). Specifically, in the loss function to train our NN (1), for the regularization parameter, we have replaced the theoretical choice of $\frac{1}{2} m \lambda \left\| \theta - \theta_0 \right\|_2^2$ ($m$ is the width of the NN) by simply $\lambda \left\| \theta \right\|_2^2$; regarding the random features of the NTK, we have replaced the theoretical choice of $\frac{1}{\sqrt{m}} \nabla h(x; \theta_t)$ by simply $\nabla h(x; \theta_t)$.

**Selection of the First Prompt.** Step $(f)$ results from the way in which the first prompt is chosen: $x_{t,1} = \arg\max_{x \in \mathcal{X}} \hat{\theta}_t^\top x$. This is analogous to the way in which our APOHF algorithm selects the first prompt: $x_{t,1} = \arg\max_{x \in \mathcal{X}} h(x; \theta_t)$.

The subsequent analysis follows from standard analysis techniques for linear dueling bandits (Bengs et al., 2022). Therefore, our strategy to select the two prompts is theoretically principled.

Note that in this section, we have provided some high-level theoretical justifications for the prompt selection strategy of our APOHF algorithm. Our prompt selection strategy can, in fact, be seen as a variant of neural dueling bandit algorithms.

