# OpenReview forum: "Prompt Optimization with Human Feedback"
_ICLR.cc/2025/Conference — Submitted to ICLR 2025_

### Official Review · Reviewer_X4tm · 2024-10-21

**Soundness:** 2
**Presentation:** 3
**Contribution:** 3
**Rating:** 6
**Confidence:** 5

**Summary:**

In existing prompt optimization work, the vast majority of methods rely on numerical scores to select better-performing prompts. However, in certain real-world tasks (e.g., text-to-image generation), using numerical scores for evaluation may not be applicable. To address this issue, the authors propose a human feedback-based prompt optimization method—APOHF. This method first collects pairs of human preference data and trains a neural network to provide latent scores aligned with human preferences. Then, combining greedy search and the method of maximizing the upper confidence bound, the authors select two prompts from multiple directions, balancing both performance and diverse prompt exploration. Experimental results show that after several iterations, APOHF can generate high-quality prompts.

**Strengths:**

1. The APOHF method proposed by the authors aims to solve the problem of prompt optimization in tasks that are difficult to evaluate using numerical scores, marking a new attempt different from previous studies.
2. APOHF outperforms other baseline methods in terms of performance.
3. The wide range of experimental types further validates the effectiveness of APOHF.
4. The authors' writing is clear, and the content of the paper is concise and easy to understand.

**Weaknesses:**

### 1. Experiment

a) From Table 6 in the appendix, I understand that for the user instruction optimization task, the authors chose two tasks, "rhymes" and "word sorting" (presumably from the BigBench dataset), which can be evaluated using numerical scores (such as accuracy), making them compatible with methods like APE, OPRO, and APO. I believe the authors should discuss this in more detail. Considering the rebuttal time constraints, You may focus on an in-depth analysis of 1 or 2 methods (such as APO, OPRO). Additionally, regarding the "sentiment" task in Table 6, why did the authors choose an initial instruction with a score of 0? For a simple instruction like "Determine the sentiment category (positive or negative) of a given sentence," a score of 0 should not occur.

b) I recommend that the authors revise the way the curves are represented in the figures. The current use of square, circular, and triangular symbols makes the figures cluttered, making it difficult to discern curve details.

### 2. Motivation
a) For prompt tasks that are difficult to evaluate with numerical scores (e.g., text-to-image generation), the optimization target of the APOHF method is individual samples (such as generating images of a garden or a street). I believe the practical significance of such single-sample optimization is limited. In everyday usage, this approach has high resource costs, making it inefficient. In industry, optimizing meta prompts holds greater practical value.

b) As I know, APOHF is not the first prompt optimization work considering human feedback. Harvard and MIT have published a similar paper named **PRompt Optimization in Multi-Step Tasks (PROMST): Integrating Human Feedback and Heuristic-based Sampling (https://arxiv.org/pdf/2402.08702)**. I think it maybe diminish the contribution of this work.

**Questions:**

1. Regarding the resource consumption of APOHF, what is the specific cost? How many API calls were made in total, and how many tokens were consumed during the experiments?

2. In Figure 3, does it show the score of the newly generated prompt after each iteration? Why doesn't APOHF exhibit score fluctuations? Logically, the scores of results generated by different prompts should vary, making it unlikely for them to remain consistent.

3. In the text-to-image task, the authors measure the quality of generated images by calculating the similarity between the generated images and the ground truth. Could this similarity metric also be used as a scoring standard for methods like APO and OPRO?

4. How many data samples does the test set include?

5. Please answer my question refer to **Weakness  Section**

---

> ### Author Response · Authors · 2024-11-25
> **Author response part 1/2**
>
> We would like to thank the reviewer for taking the time to review our paper. We also thank the reviewer for acknowledging that our proposed method is novel and effective, our experiments are extensive and our presentation is clear. We would like to address your specific questions below.
>
> > a) From Table 6 in the appendix, I understand that for the user instruction optimization task, the authors chose two tasks, "rhymes" and "word sorting" (presumably from the BigBench dataset), which can be evaluated using numerical scores (such as accuracy), making them compatible with methods like APE, OPRO, and APO. I believe the authors should discuss this in more detail. Considering the rebuttal time constraints, You may focus on an in-depth analysis of 1 or 2 methods (such as APO, OPRO).
>
> We would like to clarify that in the user instruction optimization tasks, the numerical scores are only used to simulate the human preference feedback, and are not directly used as observations. Therefore, if we compare with other methods which make use of these numerical scores (such as APE, OPRO and APO), these methods will be considered oracle methods (since they can access more information) and hence the comparisons will be unfair for our APOHF.
>
>
> > Additionally, regarding the "sentiment" task in Table 6, why did the authors choose an initial instruction with a score of 0? For a simple instruction like "Determine the sentiment category (positive or negative) of a given sentence," a score of 0 should not occur.
>
> Note that the initial score is not for the initial task description; instead, it is the prompt selected using a random initialization strategy. Specifically, since in the 0th iteration, our NN is not trained and hence we select the prompt randomly from the domain generated by the initial task description. In this case, the randomly selected prompt happens to have a score of 0.
>
> > I recommend that the authors revise the way the curves are represented in the figures. The current use of square, circular, and triangular symbols makes the figures cluttered, making it difficult to discern curve details.
>
> Thanks for pointing this out, we will make the figures clearer in our next revision.
>
> > a) For prompt tasks that are difficult to evaluate with numerical scores (e.g., text-to-image generation), the optimization target of the APOHF method is individual samples (such as generating images of a garden or a street). I believe the practical significance of such single-sample optimization is limited. In everyday usage, this approach has high resource costs, making it inefficient. In industry, optimizing meta prompts holds greater practical value.
>
> Indeed, we are facing a trade-off between the amount of resources used and the level of personalization. In our work, we aim to achieve a high level of personalization by optimizing the prompt for every user, which indeed has high resource costs. In the case of industry applications requiring low resource costs, we can extend our method to include the task information in modeling the latent score function. Specifically, instead of learning a new score function for all the prompts, we can learn how good a prompt is w.r.t. the task that we are considering. To achieve this, we can add task information to the input of our latent score model (e.g., by including the initial task description as the prefix of the input) to learn the score of the prompt w.r.t. a task. Therefore, when a new task arrives, our score function can be directly used to assess the quality of prompts w.r.t. the new task.
>
> > b) As I know, APOHF is not the first prompt optimization work considering human feedback. Harvard and MIT have published a similar paper named PRompt Optimization in Multi-Step Tasks (PROMST): Integrating Human Feedback and Heuristic-based Sampling (https://arxiv.org/pdf/2402.08702). I think it maybe diminish the contribution of this work.
>
> While the work proposed by the reviewer is relevant to our work, the problems we are studying are drastically different. Specifically, their algorithm cannot be applied in our problem settings (i.e., prompt optimization with human feedback) and hence does not diminish our contribution. Thanks for bringing this work up, we will add this discussion in our next revision.
>
> > Regarding the resource consumption of APOHF, what is the specific cost? How many API calls were made in total, and how many tokens were consumed during the experiments?
>
> We wish to point out that while these resources are important and may be costly, they are not the most expensive resources in our problem setting. Instead, human feedback is the most crucial resource (which is consistent with the problem of reinforcement learning with human feedback) and our algorithm can achieve better performance under the same number of human feedback as other baselines.

---

> ### Author Response · Authors · 2024-11-25
> **Author response part 2/2**
>
> > In Figure 3, does it show the score of the newly generated prompt after each iteration? Why doesn't APOHF exhibit score fluctuations? Logically, the scores of results generated by different prompts should vary, making it unlikely for them to remain consistent.
>
> Yes, the score in Figure 3 is the score of the first prompt selected by our algorithm (i.e., taking the argmax of the NN output as described in Line 177). Our APOHF fluctuates in the first few iterations since the NN has been updated with new data. However, as more data are collected, the input that gives the maximum NN output is stabilized and hence does not change after certain iterations. This phenomenon also exists in the baselines of Linear Dueling Bandits and DoubleTS for the same reason.
>
> > In the text-to-image task, the authors measure the quality of generated images by calculating the similarity between the generated images and the ground truth. Could this similarity metric also be used as a scoring standard for methods like APO and OPRO?
>
> We wish to point out that this similarity metric is only used for the purpose of simulating human preference feedback. All algorithms do not have access to this ground-truth label in our setting; instead, they have access to the preference feedback simulated by the score (similarity measure with the ground-truth label). Therefore, APO and OPRO can not be run in our setting since no ground-truth labels are available and thus no scoring function can be used.
>
> > How many data samples does the test set include?
>
> We have a dataset of 20 samples to evaluate the performance of the user instruction optimization as described in Line 728. For other experiments, we do not have this dataset since the performance is directly given by the similarity measure/reward model.
>
> ---
>
> Thank you again for your time and your careful feedback. We hope our clarifications could improve your opinion of our work.

---

> > ### Comment · Reviewer_X4tm · 2024-11-25
> > **Official Comment by Reviewer X4tm**
> >
> > Thank you for your clarification. I maintain my positive score. Good luck.

---

### Official Review · Reviewer_bMVs · 2024-10-31

**Soundness:** 3
**Presentation:** 3
**Contribution:** 2
**Rating:** 3
**Confidence:** 5

**Summary:**

The authors of this paper studied prompt optimization using dueling bandit. The basic idea is to ask users to provide preference feedback on responses generated by two prompts, which is directly attributed to the quality of the prompts for further selection. Experiments on prompt optimization in text-to-text and text-to-image generation tasks, and also in response optimization, demonstrate the effectiveness of the proposed solution.

**Strengths:**

1.	Prompt optimization is an important problem in LLM studies, and the paper provides a valid perspective.
2.	The clarity of the manuscript is satisfactory, which helps readers to best comprehend the technique details.

**Weaknesses:**

1.	The proposed solution is rather standard: running dueling bandits on top of ChatGPT generated rewrites of initial queries. The synergy between these two components is very weak. Ideally, the proposal of new prompts should be informed by the learnt scoring function. But the current way, the new prompts are sampled from ChatGPT independently from the scoring function.
2.	As the users’ feedback is about the response, rather than the prompt itself, the LLM that generates the responses matters. Specifically, the prompt quality is a function of generator LLM as well. This clearly limits the generality of the proposed solution: the scoring function learnt with one LLM might not work well for other LLMs. Unfortunately, there is no experiment investigating this factor.
3.	A few statements made in the paper are somehow overclaiming, for example, using a text encoder can avoid the need of a whitebox LLM, which does not seem to be advantageous, since the availability of whitebox LLM is no worse than an opensource text encoder. And I do not see what the principle is behind choosing the highest scored prompt in history as the first prompt, except it works better than randomly choosing two.

**Questions:**

1.	As the user feedback is provided to the response, instead of the prompt directly, how to we account for the variance caused by sampling from the LLM? For example, the user feels one result being better than another could be caused by LLM sampling, rather than the prompt. And I am not sure if this variance can be simply factored into the BT model.
2.	Does the learnt scoring function work across different generator LLMs? Similarly, does the learnt scoring function generalize across different tasks, e.g., question answering vs., text summarization?

---

> ### Author Response · Authors · 2024-11-25
> **Author response**
>
> We would like to thank the reviewer for taking the time to review our paper. We also thank the reviewer for acknowledging that the problem we are studying is important, and our presentation is clear. We would like to address your specific questions below.
>
>
> > Ideally, the proposal of new prompts should be informed by the learnt scoring function. But the current way, the new prompts are sampled from ChatGPT independently from the scoring function.
>
> We wish to clarify that **the sampling of new prompts (to query for human preference feedback) is indeed informed by the learned scoring function**. Specifically, our candidate prompt domain is generated by ChatGPT, and in every iteration of our algorithm, we select from the domain the next two prompts based on the prediction and the uncertainty of the prediction from the learned scoring function (shown in Line 177 and Equ. (2)). Therefore, the selection of new prompts to evaluate the human preference is indeed informed by the learned scoring function.
>
> > the prompt quality is a function of generator LLM as well. ... the scoring function learned with one LLM might not work well for other LLMs. Unfortunately, there is no experiment investigating this factor.
>
> Indeed, the prompt quality is a function of LLM and, hence, the best prompt can be determined by the LLM used here. However, we have indeed considered one application scenario (Section 4.3) where we apply our method to response selection where the quality of response is only dependent on the human and hence is transferable among different LLMs.
>
> > A few statements made in the paper are somehow overclaiming, for example, using a text encoder can avoid the need of a whitebox LLM, which does not seem to be advantageous, since the availability of whitebox LLM is no worse than an opensource text encoder.
>
> Some of the existing works (Chen et al., 2023; Hu et al., 2024; Lin et al., 2024) use the open-source whitbox LLM to obtain the last token hidden state as the representation of the prompt. However, this is very expensive since existing works use LLMs with 13B parameters (e.g., Vicuna and LLAMA), requiring a GPU with more than 15 GB VRAM to run the algorithm. However, in our case, we are using sentence-BERT which contains only 0.1B parameters and, hence, reduces the requirement of computational resources to run our algorithm.
>
> > And I do not see what the principle is behind choosing the highest scored prompt in history as the first prompt, except it works better than randomly choosing two.
>
> Note that the way to choose the first and the second prompts is **inspired by the theory** of linear dueling bandits (Bengs et al., 2022) and is hence **a principled prompt selection strategy**. Linear dueling bandits considers the case of the linear reward model and provides theoretical analysis. We have extended their selection strategies to the NN setting in the justification in Appendix C.
>
> > As the user feedback is provided to the response, instead of the prompt directly, how do we account for the variance caused by sampling from the LLM?
>
> We wish to point out that we have indeed considered the noisy setting in terms of the human preference feedback (see Lines 119-122). Specifcially, our modeling is able to take into account different types of noise, including the variance caused by sampling from the LLM.
>
> > And I am not sure if this variance can be simply factored into the BT model.
>
> Yes, it is considered. The Bernoulli distribution (from which the human preference feedback is sampled) already takes into account all noise sources from the environment including this one. On the other hand, if one would like to reduce this variance, it can be achieved by choosing a smaller temperature for the LLM generation.
>
> > Similarly, does the learnt scoring function generalize across different tasks, e.g., question answering vs., text summarization?
>
> Note that we do have experiments that learn the scoring function across different tasks, i.e., the response optimization in Sec. 4.3. In this experiment, we consider general prompts in the dataset of the Anthropic Helpfulness and Harmlessness datasets (Bai et al., 2022), which contain prompts for different tasks.
>
> On the other hand, our APOHF can be easily adapted to consider different tasks for other experiments (i.e., prompt optimization in Sec. 4.1 and Sec. 4.2) by adding the task description as a part of the input to the scoring function. Specifically, now the scoring function takes the task's information (e.g., a natural language description of the task) and the prompt as input and outputs the latent score of the prompt w.r.t. the task. By adding the task information, we can use the learned scoring function to predict the score of the prompt w.r.t. a new task by including the information of the new task in the input.
>
> ---
>
> Thank you again for your time and your careful feedback. We hope our clarifications could improve your opinion of our work.

---

> > ### Comment · Reviewer_bMVs · 2024-11-29
> >
> > I would like to thank the authors for the clarifications! But I have to point out that most of the responses seem based on wrong interpretations of my comments, which I have to further clarify.
> >
> > First, as the authors explained, the candidate prompt domain $\mathcal{X}$ is generated by ChatGPT, independent of the learnt scoring function. And this is exactly why I questioned “Ideally, the proposal of new prompts should be informed by the learnt scoring function.” Let’s think it in this way: if the pre-generated prompts are all low quality, can the dynamically learnt scoring function help us generate better ones on the fly?
> >
> > Second, “we have indeed considered one application scenario (Section 4.3)”, I am not sure why this experiment suggests the scenario “where the quality of response is only dependent on the human”. The process in APOHF is always using LLM-generated responses from a prompt to present or evaluate the prompt. Hence, the scoring function learnt from a generator LLM cannot be used to do prompt selection/optimization for another LLM; or at least, I do not this empirically verified.
> >
> > Third, “inspired by the theory of linear dueling bandits” is not a convincing argument: the problem addressed in this paper is actually best arm identification (or so-called experiment design), rather than dueling bandit, as we do not care how often the best arm (optimal prompt) is selected during learning, but whether the finally chosen one is the best arm. Hence, always chosen the currently best one as the first arm theoretically does not help best arm identification, even in the dueling feedback setting.
> >
> > Fourth, “we have indeed considered the noisy setting in terms of the human preference feedback (see Lines 119-122)”, unfortunately BT model is not omniscient, and there are a lot of factors it cannot capture. “choosing a smaller temperature for the LLM generation” could not help either, as it might amplify the bias in the LLM. This also raised a new question: can the learnt scoring function still be applicable, if the LLM used in APOHF training and deployment phases uses a different temperature, even though this is the same LLM?
> >
> > Fifth, “our APOHF can be easily adapted to consider different tasks”, the point is not whether APOHF can be used for different specific tasks, but whether a scoring function learnt for one task can be generalized across tasks. I do not think this setting was evaluated in the paper.

---

> > > ### Author Response · Authors · 2024-11-29
> > > **Author Response Part 1/2**
> > >
> > > Thank you for the prompt response and additional clarification of the questions. We wish to answer your following questions:
> > >
> > >
> > > > First, as the authors explained, the candidate prompt domain $\mathcal{X}$ is generated by ChatGPT, independent of the learned scoring function. And this is exactly why I questioned “Ideally, the proposal of new prompts should be informed by the learnt scoring function.” Let’s think it in this way: if the pre-generated prompts are all low quality, can the dynamically learnt scoring function help us generate better ones on the fly?
> > >
> > > Indeed, the generation of the prompt domain is vital to the performance since it determines the best performance we can get from prompt optimization. However, **we wish to point out that the focus of our work is not on the generation of the prompt domain but on the query efficiency of different algorithms**. There are a lot of existing works that focus on the generation of the prompt domain (Chen et al., 2023; Lin et al., 2024; Zhou et al.,2023). Our work focuses on finding the best prompt from the domain with as few human feedback queries as possible. This is also why we set the domain of the prompt to be the same among all the baselines that we compared in our work.
> > >
> > > In terms of using the learned function to help generate the prompt, we have not identified any principled approach to do this since the learned scoring function can be low-quality at the beginning of the algorithm due to insufficient preference data. However, empirically, one way to use this learned function is to generate a larger domain of prompts and use this learned function to filter out some of the low-performing prompts (e.g., by keeping the top $1k$ scored prompts from the domain of $10k$). As we mentioned previously, the improvement of prompt domain generation is not the focus of our work.
> > >
> > > > Second, “we have indeed considered one application scenario (Section 4.3)”, I am not sure why this experiment suggests the scenario “where the quality of response is only dependent on the human”. The process in APOHF is always using LLM-generated responses from a prompt to present or evaluate the prompt. Hence, the scoring function learnt from a generator LLM cannot be used to do prompt selection/optimization for another LLM; or at least, I do not this empirically verified.
> > >
> > > We wish to point out that the quality of the response in the experimental setting of Sec. 4.3 is not dependent on the LLM that is used. Specifically, Sec. 4.3 studies the selection of the response instead of the prompt. Therefore, instead of selecting the best prompt from the domain of the prompt, we select the best response for a prompt from the domain of the responses. Our algorithm first generates a domain of response from the LLM by sampling multiple times from the LLM and asking humans to provide preference feedback on the response. In this case, our reward function takes the concatenation of the prompt and response as input (instead of only prompt) and outputs the score of the response w.r.t. the prompt provided (i.e., the same way as the reward function in RLHF which is not dependent on the LLM that generates the response). Consequently, this score is not a function of the LLM used but solely a function of human preference and hence can be used in different LLMs.
> > >
> > >
> > >
> > >
> > > > Third, “inspired by the theory of linear dueling bandits” is not a convincing argument: the problem addressed in this paper is actually best arm identification (or so-called experiment design), rather than dueling bandit, as we do not care how often the best arm (optimal prompt) is selected during learning, but whether the finally chosen one is the best arm. Hence, always chosen the currently best one as the first arm theoretically does not help best arm identification, even in the dueling feedback setting.
> > >
> > > Previous works (Chen et al., 2023; Lin et al., 2024; [1]) have discussed and empirically shown that in practice, regret minimization algorithms (in contrast to best arm identification algorithms) usually achieve better performances in solving black-box optimization problems, such as prompt optimization. Therefore, these existing works justify the use of the acquisition function from dueling bandit (i.e., regret minimization algorithm). This is also corroborated by the strong empirical performances of our algorithm in our experiments.
> > >
> > > [1] Wu, Z., Lin, X., Dai, Z., Hu, W., Shu, Y., Ng, S. K., ... & Low, B. K. H. (2024). Prompt Optimization with EASE? Efficient Ordering-aware Automated Selection of Exemplars. NeurIPS 2024.

---

> > > ### Author Response · Authors · 2024-11-29
> > > **Author Response Part 2/2**
> > >
> > > > Fourth, “we have indeed considered the noisy setting in terms of the human preference feedback (see Lines 119-122)”, unfortunately BT model is not omniscient, and there are a lot of factors it cannot capture. “choosing a smaller temperature for the LLM generation” could not help either, as it might amplify the bias in the LLM. This also raised a new question: can the learnt scoring function still be applicable, if the LLM used in APOHF training and deployment phases uses a different temperature, even though this is the same LLM?
> > >
> > > As we have mentioned previously, the BT model is commonly used in the existing works and provides a decent noise coverage. Indeed, some of the noise may not be covered by the BT model; however, this is not the main focus of our paper. Finding a better preference modeling approach to consider all kinds of noise can be a significant contribution that can be studied as a separate work. Our focus is to design a query-efficient POHF algorithm to identify the best prompt using human feedback.
> > >
> > > Regarding the temperature, indeed, a reward function learned with a specific temperature will be different from the one learned with another temperature in the setting of Sec. 4.1 and Sec. 4.2. However, as we mentioned previously, the reward function is transferable for different LLMs in the setting of Sec. 4.3. Therefore, our approach is still useful when the temperature changes in the setting of Sec. 4.3.
> > >
> > >
> > >
> > >
> > > > Fifth, “our APOHF can be easily adapted to consider different tasks”, the point is not whether APOHF can be used for different specific tasks, but whether a scoring function learnt for one task can be generalized across tasks. I do not think this setting was evaluated in the paper.
> > >
> > > We wish to point out that in our previous response, we are referring to the case that **the learned scoring function from some tasks can be generalized to other tasks without new preference feedback in the setting of and with a minor modification on Sec. 4.1&4.2**.
> > >
> > > Specifically, in the experiment of Sec. 4.3, we consider general prompts in the dataset that contain prompts for different tasks, and we learn one single reward function with the input of concatenation of prompt and response. In that case, the score (given by one single model) evaluates the quality of the response w.r.t. each prompt that is from different tasks. Here are 3 examples from 3 tasks in this dataset:
> > >
> > >
> > > Example 1: Can you check this text for any grammar or spelling mistakes ...
> > >
> > > Example 2: Tell me all about fracking, or oil extraction, or oil drilling.
> > >
> > > Example 3: Please summarize the plot of the novel Blood Meridian in the style of a valley girl.
> > >
> > > Example 1 is the task of grammar correction, Example 2 is the task of factual QA, and Example 3 is the task of summarization. These prompts are evidently from different tasks, and we use the same reward function to model them in the setting of Sec. 4.3. Therefore, the reward function can be generalized on the unseen tasks in the test dataset where we evaluate our performance.
> > >
> > >
> > > On the other hand, in the setting of Sec. 4.1 and 4.2, our algorithm can be easily adapted to consider different tasks using one single reward function by adding the task description as a part of the input to the scoring function. Specifically, now the scoring function takes the task's information (e.g., a natural language description of the task) and the prompt as input and outputs the latent score of the prompt w.r.t. the task. By adding the task information, we can use the learned scoring function to predict the score of the prompt w.r.t. a new task by including the information of the new task in the input.
> > >
> > > ---
> > >
> > > Thank you again for your time and your careful feedback. We hope our clarifications could improve your opinion of our work.

---

### Official Review · Reviewer_Y27t · 2024-11-02

**Soundness:** 1
**Presentation:** 2
**Contribution:** 1
**Rating:** 3
**Confidence:** 4

**Summary:**

This paper mainly studies the prompt optimization problem. In specific, the main motivation coms from that previous PO methods usually require the availability of a numeric score to assess the quality of every prompt. This score is difficult or sometime unable to get in the situation when human interact with a black-box LLM system. Therefore, the authors claimed that preference feedback is more feasible given the mentioned human-llm interaction scenario. The authors thus proposed a method named APOHF for prompt optimization with human feedback. Some results show that this method can find a good prompt with several preference feedbacks used.

**Strengths:**

The paper is well written and easy to follow
The motivation is clear compared to previous method. [which may not be practical as per the weakness part]

**Weaknesses:**

The main weakness is the prompt optimization from human feedback itself may not be practical. Let me explain the reasons.
1) The advantage or the main efforts of LLM is to improve its instruction following abilities, I.e., to cover as many prompts as possible. Therefore, when human in the loop, the key efforts should be in the zero-shot/few-shot abilities. If you expect the user to give feedback to the prompt provided by him/her-self, it may negatively impact the feasible application of this LLM product. How does your approach complement efforts to improve zero-shot/few-shot abilities of LLMs? Are there specific scenarios where human feedback on prompts provides unique value, even as LLMs improve their general instruction-following capabilities?


2) Prompt optimization itself is important for sure, but the efforts on making it automatic are more useful. For instance, as many APO did, LLM can improve the prompt itself by self-reflecting (e.g., reasoning the errors and self-refining current prompts). How does your method compare to or complement automated prompt optimization techniques like self-reflection? Are there potential advantages to incorporating human feedback alongside automated methods?

3) If let’s say we need human preference, we can directly train a reward model. Given the generalization ability of the reward model, we can directly sample the best response with requiring the human-preference during inference. How does your approach compare to training a reward model on human preferences? Could you discuss the potential advantages or disadvantages of your method compared to using a pre-trained reward model?

In addition, the experiments can be done on more types of tasks, such as math, coding, role-play, etc.

**Questions:**

Most of questions have been listed in the weakness part.

Another general question or suggestion would be: Could the author please raise up several real examples that can show the scenarios where the proposed method will be feasible to apply?

---

> ### Author Response · Authors · 2024-11-25
> **Author response part 1/2**
>
> We would like to thank the reviewer for taking the time to review our paper. We also thank the reviewer for acknowledging that our paper is well written and our motivation is clear. We would like to address your specific questions below.
>
>
> > The advantage or the main efforts of LLM is to improve its instruction following abilities, I.e., to cover as many prompts as possible. Therefore, when human in the loop, the key efforts should be in the zero-shot/few-shot abilities. If you expect the user to give feedback to the prompt provided by him/her-self, it may negatively impact the feasible application of this LLM product. How does your approach complement efforts to improve zero-shot/few-shot abilities of LLMs? Are there specific scenarios where human feedback on prompts provides unique value, even as LLMs improve their general instruction-following capabilities?
>
> **Indeed, the LLM's general instruction-following abilities are important. However, our algorithm provides unique values in personalization of users' preference, which is also of paramount importance for users.** Specifically, different users have different preferences on the responses from the LLMs. For example, Table 5 shows different responses from the LLM when asked to give travel advice; some users may prefer general advice, while others may prefer more detailed advice. Therefore, our work aims to find the best responses for different users respectively (i.e., personalization). Note that this personalization does not conflict with improving instruction following abilities. In fact, these are two orthogonal aspects that will improve users' experience.
>
> On the other hand, **our approach can also be applied to zero-shot/few-shot scenarios.** If we understand correctly, "zero-shot/few-shot abilities" refers to finding the most preferred prompt/response with no or minimal human feedback. This ability can be achieved by adding user features as the context in our latent score prediction model. Specifically, the user feature (e.g., demographic data) can be added as an input to the model (i.e., combined with the prompt information) to predict the latent score of the prompt w.r.t. a specific user. Consequently, when we have a new user, we can still use the latent score function learned previously to predict the new user's preference without additional human feedback.
>
>
> > Prompt optimization itself is important for sure, but the efforts on making it automatic are more useful. For instance, as many APO did, LLM can improve the prompt itself by self-reflecting (e.g., reasoning the errors and self-refining current prompts). How does your method compare to or complement automated prompt optimization techniques like self-reflection? Are there potential advantages to incorporating human feedback alongside automated methods?
>
> Note that **we are not trying to improve previous methods (e.g., self-reflecting methods) by incorporating human feedback. Instead, we are solving a new problem, which previous works can not work (i.e., only human feedback is available)**. Specifically, previous works (Zhou et al. (2023); Yang et al. (2024); Lin et al. (2024); Hu et al. (2024)) all require a scoring method for the response to find the best prompt. However, this scoring method is not available (e.g., no validation dataset available) in most real applications, rendering these existing methods inapplicable. Our work is the first to propose to optimize the prompt based on human feedback when no scoring method is available.
>
> On the other hand, as we mentioned previously, different users have different preferences and hence different best prompts. However, the existing self-reflecting method does not take each user's personalized preference into consideration and hence can only give the same best prompt for different users, which is undesirable in real applications.

---

> ### Author Response · Authors · 2024-11-25
> **Author response part 2/2**
>
> > If let’s say we need human preference, we can directly train a reward model. Given the generalization ability of the reward model, we can directly sample the best response with requiring the human-preference during inference. How does your approach compare to training a reward model on human preferences? Could you discuss the potential advantages or disadvantages of your method compared to using a pre-trained reward model?
>
> Indeed a pre-trained reward model can be used to predict the general preference (i.e., preference of the crowd), however, our method focuses on the preference of a specific user (i.e., personalization). Therefore, instead of using a general reward model, we need a reward model for different users. Our APOHF provides an efficient way to actively collect human feedback to train a reward model for a specific user. Therefore, the advantage of our method is that we can learn the preferences of different users (i.e., personalization) while the pre-trained reward function does not take this into account. The disadvantage of our method is that we require new human feedback from new users in order to determine the best prompt, which can be addressed by adding user features in the latent score prediction model as we have discussed earlier.
>
> >  Could the author please raise up several real examples that can show the scenarios where the proposed method will be feasible to apply?
>
> We wish to point out that our experimental settings resemble real application scenarios, especially the prompt optimization for generating ideal images (i.e., Sec. 4.2) and response optimization (i.e., Sec. 4.3). Specifically, in Sec. 4.2, we consider the real application of optimizing the prompt for generating ideal images from text-to-image models (i.e., DALLE model from OpenAI) in which the user provides a prompt (e.g., the prompt about generating a garden image in Table 3) and wish to get an image that is most similar to what he/she has in mind. In Sec. 4.3, we optimize the response for general user prompts (e.g., the prompt asking for travel advice in Table 5). These are real-wold examples and we have shown that our method is feasible to apply in these applications.
>
>
> ---
>
> Thank you again for your time and your careful feedback. We hope our clarifications could improve your opinion of our work.

---

### Official Review · Reviewer_WviX · 2024-11-03

**Soundness:** 2
**Presentation:** 3
**Contribution:** 3
**Rating:** 6
**Confidence:** 5

**Summary:**

This paper mentions a common issue of black-box LLM usages, where it is unavailable to automatically obtain a quality score for a given prompt, thus causing the difficulty of prompt optimization. Assuming that only human’s preference feedback is reliable, the authors propose APOHF, a framework to perform prompt optimization with human feedback. The proposed method aims to determine a good prompt via a prompt selection algorithm inspired by Dueling Bandits, based on a neural network performance predictor trained with the information from a small number of human feedback instances. The authors demonstrate quality improvement throughout iterations of the optimization. Overall, the experiment results show that the proposed framework is effective to obtain appropriate prompts with human feedback.

**Strengths:**

-  Inspired by the concept of RLHF and Dueling Bandits, the authors propose a prompt optimization framework leveraging human feedback. The experiment results show that the proposed method is indeed effective to a certain extent with actual demonstrations.

-  With the Bandit-fashioned prompt selection strategy, powered by a trained NN (MLP) model as the performance predictor, the proposed framework determines the prompt efficiently in terms of the number of required human feedback instances.

-  In the appendix, the authors also provide a brief coverage of theoretical explanations to justify the proposed prompt selection strategy.

**Weaknesses:**

-   Limited Comparison with Prompt Optimization Baselines:
The authors' decision to exclude comparisons with state-of-the-art prompt optimization methods (e.g. TextGrad), citing the lack of a scoring method, may be overly restrictive. While direct human scoring might not be feasible, surrogate evaluation metrics (e.g., embedding-similarity between LLM output and ground truth) could serve as viable alternatives for these comparisons. Given the superior performance of existing prompt optimization methods across various tasks, their inclusion in the performance comparisons would provide valuable context and a more comprehensive evaluation of APOHF's effectiveness relative to the current state of the art.

- Dependency on Neural Network Models for Prompt Selection:
The proposed framework's prompt-pair selection strategy heavily relies on a trained neural network (MLP) model for performance prediction based on input embeddings. This approach introduces two potential sources of variability:
a) The choice of embedding model
b) The architecture and training of the performance predictor (MLP)
Both components may significantly influence the accuracy of performance predictions, potentially leading to substantial variations in the final LLM performance. A more robust analysis of these components' impact on the overall framework would strengthen the paper's credibility.

- Ambiguity in Prompt Domain Generation:
The process of generating the "discrete domain of prompts X" lacks sufficient detail. While the authors mention using a powerful LLM (e.g., ChatGPT) for prompt generation via in-context learning, the specifics of this crucial step remain unclear. Moreover, the paper lacks experimental analysis demonstrating how different prompt domain generation methods affect APOHF's overall performance. This omission limits the reader's ability to fully assess the method's robustness and generalizability.

**Questions:**

see weaknesses

---

> ### Author Response · Authors · 2024-11-25
> **Author response**
>
> We would like to thank the reviewer for taking the time to review our paper. We also thank the reviewer for acknowledging that our proposed method is theoretically justified and effective. We would like to address your specific questions below.
>
>
> > Limited Comparison with Prompt Optimization Baselines: The authors' decision to exclude comparisons with state-of-the-art prompt optimization methods (e.g. TextGrad), citing the lack of a scoring method, may be overly restrictive. While direct human scoring might not be feasible, surrogate evaluation metrics (e.g., embedding similarity between LLM output and ground truth) could serve as viable alternatives for these comparisons. Given the superior performance of existing prompt optimization methods across various tasks, their inclusion in the performance comparisons would provide valuable context and a more comprehensive evaluation of APOHF's effectiveness relative to the current state of the art.
>
> We wish to clarify that we do not have the ground-truth label available to our algorithm in this case; therefore, the "embedding-similarity between LLM output and ground truth" proposed here is not applicable in our case. Specifically, our target application scenarios are those in which we do not have the ground-truth label for us to compute scoring metrics but only human preference feedback. Therefore, the ground-truth label is unavailable in our case, rendering the existing methods that require ground-truth label for the scoring function not applicable here.
>
> > Dependency on Neural Network Models for Prompt Selection: The proposed framework's prompt-pair selection strategy heavily relies on a trained neural network (MLP) model for performance prediction based on input embeddings. This approach introduces two potential sources of variability: a) The choice of embedding model b) The architecture and training of the performance predictor (MLP) Both components may significantly influence the accuracy of performance predictions, potentially leading to substantial variations in the final LLM performance. A more robust analysis of these components' impact on the overall framework would strengthen the paper's credibility.
>
> Indeed, more ablation studies on the choice of embedding model and MLP architecture can provide more insight. However, the previous work of (Hu et al. 2024), which has also adopted pre-trained embedding and neural networks for score prediction, has shown their influences in the context of prompt optimization. Due to time and resource constraints, we did not repeat similar experiments.
>
> On the other hand, our APOHF outperforms the other baselines in all scenarios with a fixed embedding and MLP architecture (without hyper-parameter tuning), which we think demonstrates the robustness of our method.
>
> > Ambiguity in Prompt Domain Generation: The process of generating the "discrete domain of prompts X" lacks sufficient detail. While the authors mention using a powerful LLM (e.g., ChatGPT) for prompt generation via in-context learning, the specifics of this crucial step remain unclear. Moreover, the paper lacks experimental analysis demonstrating how different prompt domain generation methods affect APOHF's overall performance. This omission limits the reader's ability to fully assess the method's robustness and generalizability.
>
> We have in fact described the detailed steps for generating the prompt domain for all 3 experimental settings we consider. Specifically, for user instruction optimization, we have described the detailed template used to generate the prompt in Example Query 1 in Appendix A. For prompt optimization in text-to-image models, we have provided the templates to generate the prompt in Example Query 2 & 3 in Appendix B with detailed steps in Lines 733-751. For response optimization, we generate the domain of response using steps in Lines 753-755. We believe that these descriptions provide sufficient details for reproducibility.
>
> On the other hand, the way of generating prompt domains is not the focus of our work; existing works have provided extensive study on this, such as (Hu et al. 2024). Our focus is on selecting prompt pairs to learn the latent scoring function efficiently. Hence, we use the same domain of prompts/responses for fair comparison.
>
> ---
>
> Thank you again for your time and your careful feedback. We hope our clarifications could improve your opinion of our work.

---

> > ### Comment · Reviewer_WviX · 2024-12-03
> >
> > The authors have addressed some of my concerns and I have raised my score. Nevertheless, I still feel comparing with the STOA is a must to justify the setup is more meaningful comparing to the conventional PO framework.

---

### Official Review · Reviewer_jWbb · 2024-11-04

**Soundness:** 3
**Presentation:** 3
**Contribution:** 3
**Rating:** 5
**Confidence:** 4

**Summary:**

This paper introduces APOHF, for optimizing prompts for large language models (LLMs) using only human preference feedback rather than numeric scores. APOHF iteratively selects prompt pairs for user comparison, training a neural network to predict each prompt’s utility based on feedback. The algorithm selects prompts by combining utility prediction with an exploration-exploitation approach inspired by dueling bandits. Applied across tasks like instruction optimization and prompt tuning for text-to-image models, APOHF demonstrates more efficient prompt selection compared to baseline methods under limited feedback conditions.

**Strengths:**

* This paper presents a novel approach to prompt optimization by using human preference feedback alone, which is suitable for black-box LLMs.
* This paper is generally clear in its method description, though certain theoretical justifications could be expanded for a more rigorous understanding of APOHF’s design choices.
* The algorithm design balances prompt utility prediction with exploration, showing reliable performance across varied tasks.

**Weaknesses:**

* The method employs the Bradley-Terry-Luce (BTL) model to represent human preference feedback, which assumes consistency and transitivity in user preferences. This may oversimplify human feedback, especially in real-world applications where user preferences can be inconsistent or influenced by context. The reliance on binary feedback might also limit the granularity of information available to the model, potentially leading to suboptimal prompt choices.
* The method relies on a user-provided initial task description to generate the prompt domain, assuming that these initial examples are representative of the task requirements. This dependency introduces the potential for bias if the initial examples do not capture the full scope of the task or if the user’s interpretation is inconsistent with the intended outcomes. This reliance can constrain the model's flexibility and lead to prompts that are effective only in limited or narrowly defined scenarios.
* APOHF presumes that user preferences are consistent and relevant across multiple iterations, assuming stability in what constitutes an optimal prompt. However, in complex, open-ended tasks or tasks that evolve over time, user preferences may shift, and certain prompts that were optimal initially may no longer be relevant. This assumption limits the model's adaptability to dynamic contexts, reducing its applicability in real-world tasks where user expectations or task goals may evolve.

**Questions:**

* Have the authors examined how noise in user feedback impacts the accuracy of the prompt selection? Is the model robust to feedback inconsistencies or context-dependency in user preferences?
* Is there a way to iteratively refine or expand the prompt domain based on user feedback, to counteract biases introduced by an incomplete initial task description?
* Has the algorithm been tested in settings where user preferences or task requirements change over time? If so, how does APOHF handle shifts in user preferences?

---

> ### Author Response · Authors · 2024-11-25
> **Author response**
>
> We would like to thank the reviewer for taking the time to review our paper. We also thank the reviewer for acknowledging that our approach is novel, our presentation is clear and the performance of our approach is reliable. We would like to address your specific questions below.
>
> > The method employs the Bradley-Terry-Luce (BTL) model to represent human preference feedback, which assumes consistency and transitivity in user preferences. This may oversimplify human feedback, especially in real-world applications where user preferences can be inconsistent or influenced by context. The reliance on binary feedback might also limit the granularity of information available to the model, potentially leading to suboptimal prompt choices.
>
> We wish to point out that the BTL model is widely used in previous works in RLHF (Bai et al., 2022, Ouyang et al. 2022) and has been shown to be able to handle human preference modeling very well empirically. Note that the problem of inconsistency in human preference is not unique to our work, and the BTL model can actually handle this inconsistency well by considering noisy observations (as we mentioned in Lines 119-120). Therefore, we believe that BTL is a good design choice here.
>
> > The method relies on a user-provided initial task description to generate the prompt domain, assuming that these initial examples are representative of the task requirements. This dependency introduces the potential for bias if the initial examples do not capture the full scope of the task or if the user’s interpretation is inconsistent with the intended outcomes. This reliance can constrain the model's flexibility and lead to prompts that are effective only in limited or narrowly defined scenarios.
>
> Indeed, if the initial task description does not capture the full scope of the task, the generated prompt candidates will be affected. However, we wish to point out that this requirement is not unique to us and is also applicable to numerous previous works (Zhou et al. 2023; Yang et al. 2024; Lin et al. 2024; Hu et al. 2024). In addition, **our work has already proposed some methods to have a better coverage of the full scope of the task.** Specifically, for the text-to-image prompt optimization task, we proposed ways to modify the content of the initial information by adding/deleting information (see Example Query 2 & 3 in Appendix A) from the prompt such that the generated prompt candidates space will have a decent coverage of neighbors of the initial task description. For the instruction induction task, we proposed to use multiple subsets of exemplars (i.e., randomly sample 5 out of 100 each time as described in Lines 254-256) as the initial task description such that we will have some subsets of exemplars that are more representative of the task. Therefore, our approach has taken the bias of the initial task description into account and has addressed this problem to some extent.
>
> > APOHF presumes that user preferences are consistent and relevant across multiple iterations, assuming stability in what constitutes an optimal prompt. However, in complex, open-ended tasks or tasks that evolve over time, user preferences may shift, and certain prompts that were optimal initially may no longer be relevant. This assumption limits the model's adaptability to dynamic contexts, reducing its applicability in real-world tasks where user expectations or task goals may evolve.
>
> We would like to kindly point out that our work is the first to study prompt optimization with human feedback. Time-varying human preference is a relevant and important problem that can be studied separately in future work and is beyond the scope of our current work. We plan to expand our approach in future work to time-varying preferences based on non-stationary dueling bandits.
>
> > Have the authors examined how noise in user feedback impacts the accuracy of the prompt selection?
>
> Yes, we have investigated the impact of noise on the performance of the prompt in Fig. 8. Specifically, $s$ controls the noise level, a larger $s$ indicates a larger noise. We increase $s$ from $0.1$ (low noise) to $1000$ (high noise), and the performance of the best prompt identified by all methods decreases. However, our method still performs decently well compared to the other baselines.
>
> ---
>
> Thank you again for your time and your careful feedback. We hope our clarifications could improve your opinion of our work.

---

### Meta-Review · Area_Chair_ZmXR · 2024-12-18

**Metareview:**

This paper explores a novel approach to prompt optimization by leveraging human preferences instead of numerical scores as feedback. It employs a dueling bandit algorithm to efficiently select pairs of prompts for comparison in each iteration.

While the paper is well-written and presents a unique concept in prompt optimization, reviewers have raised concerns regarding its practicality. Specifically, they suggest further investigation into: 1) Prompt domain generation methods: A deeper analysis of how the initial pool of prompts is generated; 2) Embedding model and MLP: Ablation studies to understand the impact of different embedding models and multilayer perceptrons on performance; 3) Comparison with other methods: A comparative analysis with existing prompt optimization algorithms to demonstrate the effectiveness of the proposed approach.

In essence, I would recommend strengthening the paper by addressing these practical considerations and providing a more comprehensive evaluation of the proposed method.

**Additional Comments On Reviewer Discussion:**

The reviewers' primary concerns centered on the need for ablation studies and comparisons with existing prompt optimization algorithms. However, the authors failed to adequately address these concerns in their rebuttal.

---

### Decision · Program_Chairs · 2025-01-22

Reject